# Human confidence judgments reflect reliability-based hierarchical integration of contextual information

Philipp Schustek [1,3]*, Alexandre Hyafil [1,2,3]* & Rubén Moreno-Bote[1]

Our immediate observations must be supplemented with contextual information to resolve ambiguities. However, the context is often ambiguous too, and thus it should be inferred itself to guide behavior. Here, we introduce a novel hierarchical task (airplane task) in which participants should infer a higher-level, contextual variable to inform probabilistic inference about a hidden dependent variable at a lower level. By controlling the reliability of past sensory evidence through varying the sample size of the observations, we find that humans estimate the reliability of the context and combine it with current sensory uncertainty to inform their confidence reports. Behavior closely follows inference by probabilistic message passing between latent variables across hierarchical state representations. Commonly reported inferential fallacies, such as sample size insensitivity, are not present, and neither did participants appear to rely on simple heuristics. Our results reveal uncertainty-sensitive integration of information at different hierarchical levels and temporal scales.

[1] Center for Brain and Cognition & Dept. of Information and Communication Technologies, Pompeu Fabra University, 08005 Barcelona, Spain. [2] Center for Research in Mathematics, Campus UAB Edifici C, 08193 Bellaterra, Barcelona, Spain. [3] These authors contributed equally: Philipp Schustek, Alexandre Hyafil. *email: philipp.schustek@gmail.com; alexandre.hyafil@gmail.com

As sensory evidence is inherently ambiguous, it needs to be integrated with contextual information to minimize the uncertainty of our perception of the world and thus allow for successful behavior. Suppose that we observe just a few passengers exiting an airplane at an airport whose city hosts a soccer final. If we find that four of them are supporters of the red team and two support the blue team, we might conclude that there were more supporters of the red team in the airplane. This inference, based on incomplete sensory evidence, can be improved by contextual information. For instance, there might be many more blue than red supporters in the world. Then, despite our initial observation, we might want to revise our inference and rather conclude, based on the context, that the airplane carried more blue than red supporters.

While in the previous example context was certain and by itself able to resolve observational ambiguity, contextual information is very often ambiguous. For instance, we might just know that there is an event in the city that attracts more of a certain type of people, but we do not know which type. Extending our example, we would first need to infer the context (whether the event attracts more people of the red or blue type) by observing samples of passengers leaving several airplanes. By using the inferred context, we can better estimate whether the next plane carries more of one type of people given only on a small sample of its passengers. Thus, in real-life, both observations and context commonly provide incomplete information about a behaviorally relevant latent variable. In these cases, inference should be based on probabilistic representations of both observational and contextual information[1–6].

Indeed, recent work has shown that humans can track a contextual binary variable embedded in noise that partially informs about what specific actions need to be performed to obtain reward[7]. Additionally, humans can infer the transition probability between two stimuli where the transition probability itself undergoes unexpected changes, defining a partially observable context[8]. These results and other studies suggest that a refined form of uncertainty representation is held at several hierarchical levels by the brain[9–14]. However, in this previous research, the reliability of the context has rarely been manipulated directly and independently[15] from the reliability of the current observation[1,7,8]. Therefore, it is unclear up to what degree contextual inference reflects its uncertainty and interacts with the inferred reliability of the current observation as it would be expected from representing it with a joint probability distribution over both observations and context. While some effects predicted by hierarchical probabilistic inference have been previously reported in isolation, no study has —to our knowledge—thoroughly assessed a body of behavioral predictions of hierarchical probabilistic inference and tested them against alternative heuristics model.

To address the above question, we developed a reliability-based hierarchical integration task that allows us to directly control reliability in order to evidence characteristic patterns of probabilistic inference. Our task was intuitively framed to our participants using the analogy of flight arrivals to an airport whose city hosts an event, rather than relying on an abstract or mathematical description of the dependencies between the latent variables. The goal was to decide whether the flight just landed carried more passengers of the red or blue type based on the observation of only a small sample of passengers leaving the airplane, and to report the confidence in that decision. However, as the event is known to tend to attract more of either of the two types of passengers, knowledge of this context, if inferred correctly, would be useful to solve the task. The crucial ingredient of our task is that inference of the context is based on the observation of small samples of passengers exiting previously arrived planes, making the context partially, but not fully, observable. By manipulating both the tendency and the sample size, we can control the reliability of previous observations upon which inference about the context should be based. Overall, this task structure creates hierarchical dependencies among latent variables that should be resolved by bottom-up (inferring the context from previous observations) and top-down message passing (inferring the current state by combining current observations with the inferred context)[6].

We find that participants can track and use the inferred reliability of previous observations suggesting that they build a probabilistic representation of the context. The inferred context is integrated with the current observations to guide decisions and confidence judgments about the value of a latent variable at a lower hierarchical level. Decision confidence is found to closely correspond to the actual accuracy of making correct decisions. As a clear signature of probabilistic inference over the context, we find that the sample size of previous observations is used by our participants to infer the reliability of the context. This in turn has a strong effect on decision confidence of a lower-level variable that depends on the context. The observed behavior in our participants eludes previously reported biases in judgments and decision making[16], such as sample size insensitivity[17–19], and also resists explanations based on simpler heuristics[20,21]. Overall, all the reported effects in both tasks are consistent, quantitatively and qualitatively, with the optimal inference model. Thus, our results support the view that humans may form mental representations akin to hierarchical graphs[22] that support reliability-based inference to guide confidence estimates of our decisions.

## Results

**The airplane task probes inference of latent variable**. We designed two experiments to test whether humans can use the reliability of contextual information to guide decisions and confidence judgments about a latent variable at a lower hierarchical level. While in some previous studies, instructions to the participants were quite abstract and often appealed to mathematical terms[21], here we attempted to facilitate understanding of the complex relationships of the task variables by instructing participants in intuitive and naturalistic terms. Thus, we described the task to our participants by using the analogy of airplanes arriving at an airport whose unknown passenger proportions were to be estimated. In the first experiment (Experiment 1), the context is neutral and stable across all the trials encompassing the session, while in the second experiment (Experiment 2) context varies across blocks of a few trials but remains constant within each block. We instructed our participants that the context consists of a tendency of the encountered airplanes to carry more passengers of either of the two types. Formally, Experiment 1 corresponds to the classical urn problem with unknown fractions of red and blue balls, and Experiment 2 corresponds to a hierarchical extension where the urns are themselves correlated and partially observable (see Methods). As no feedback was given that instructed our participants how they ought to make their confidence reports, the experiments probe their internal capacity to estimate uncertainties.

**The effects of sample size on confidence reports**. In Experiment 1, participants were told that the airplanes arriving to an airport carry both blue- and red-type passengers, in an unknown proportion, and that these proportions would be uncorrelated from one plane to the next. Thus, in this case, no context was assumed that would make our participants believe that the passenger proportions across consecutives planes would be interdependent. After observing a small sample of passengers randomly exiting the plane, displayed as red and blue filled circles on the screen

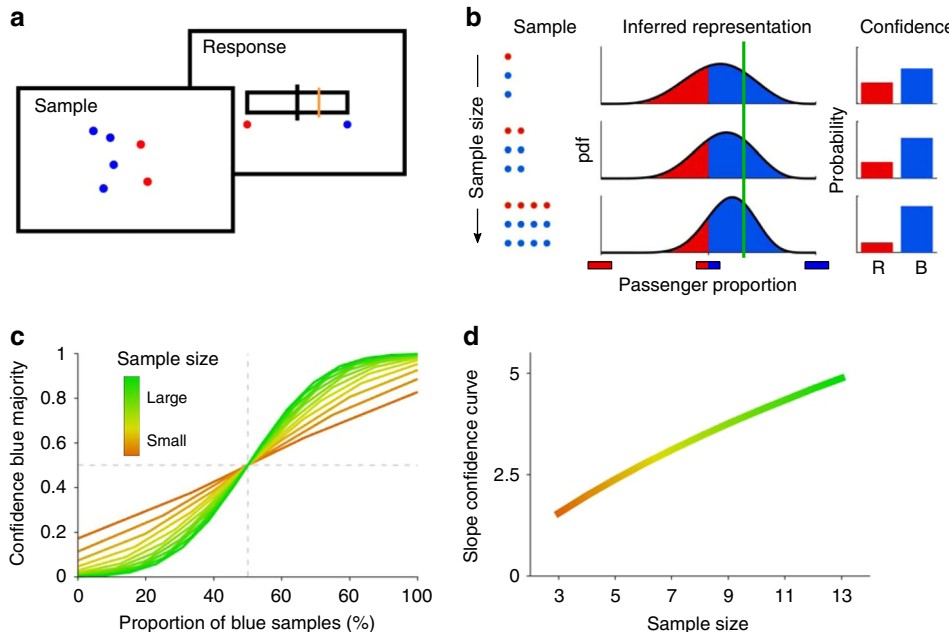

**Fig. 1** Posterior-based confidence features sample size effects **a** Task: The colored dots (sample) represent two kinds of passengers (blue and red) that disembarked a very large airplane. The participants are subsequently asked to report the confidence in their decision that the airplane carried more blue or red passengers (blue majority) by horizontally moving the cursor line (orange). In this case, because the sample suggests a blue majority, the response cursor should be on the right. **b** Sample size increases posterior-based confidence in a blue majority suggested by the blue majority of the sample. Confidence (right) is computed as expected accuracy from the area under the curve for the inferred proportion (middle) from the observed sample (left). Although the proportion of blue passengers (green line, middle) is the same for all three samples (rows), the inferred distribution depends on sample size. The larger the confidence, the closer the response line on the previous panel should be to the rightmost border. **c** Confidence in blue majority should increase with the proportion (%) of blue samples for all sample sizes, but it does so with a higher slope for larger sample sizes (color coded). **d** Consequently, the slope parameter of fitted sigmoidal functions increases with sample size.

(Fig. 1a, first frame), participants were asked to report both whether the airplane carried more blue or red passengers, i.e., its passenger majority, and their confidence in this decision by moving a line along a horizontal bar (second frame). Importantly, there was no direct feedback about normative confidence reports: participants received a binary feedback after each response, i.e., whether they correctly identified the latent passenger majority. In addition, indirect feedback was provided at regular pauses every five trials through some aggregated performance score based on the ideal observer which was solely intended to maintain our participants engaged in the task. While such feedback could in principle be marginally used to adapt one's responses, participants did not seem to modify their responses accordingly: first, feedback was hardly indicative of the optimal policy (see Methods); second, participants performed the task well from trial one and did not improve over time (see Supplementary Fig. 6a).

An ideal observer (Fig. 1b) should infer a distribution over an airplane's proportion of blue (or, equivalently, red) passengers based on the observed proportion of blue passengers and the sample size. The proportion of blue samples (passengers), called "sample proportion", is computed as $N_B/N$, where $N_B$ ($N_R$) is the number of observed blue (red) passengers, respectively, and $N = N_B + N_R$ is the sample size. The inferred distribution over passenger proportions concentrates around passenger proportions suggested by the sample (Fig. 1b, green vertical line)[17], and its width reduces the larger the sample size is. The decision whether the majority is blue or red is uniquely based on the proportion of blue samples, but the confidence report should be based on both the sample proportion and the sample size. Specifically, in this example, decision confidence of the ideal observer is the belief that the majority is blue, which equals the area under the distribution summing up the probability of all

possible blue passenger proportions that are larger than one half[23,24] (Fig. 1c, d). The result is that confidence in a blue majority increases with sample size because the distribution is more concentrated around the observed proportion of blue passengers. More generally, a central feature of probabilistic inference is sample size dependence, which here magnifies the confidence in the airplane majority that is suggested by the sample proportion.

We tested whether human participants ($n = 24$) obeyed this critical pattern or whether they neglected size[17,19]. Confidence in a blue majority was found to increase with the proportion of blue samples. As predicted, this increase was larger the larger the sample size is (Pearson correlation, pooled across participants, $\rho = 0.31$, $p = 4.08 \times 10^{-6}$) (Fig. 2a, b). These results were found for most of our participants individually (21 out of 24; permutation test, $p < 0.05$; Supplementary Fig. 2). Consistently, confidence judgments were highly predictive of the probability that the chosen majority was correct (Pearson correlation, $\rho = 0.81$, $p = 1.27 \times 10^{-45}$, see Supplementary Fig. 1 for details), suggesting that participants performed the task well and gave confidence reports that follow from an internal measure of uncertainty.

To further confirm that sample size was an important feature of our participants' confidence reports, we performed a model comparison in which we contrasted the optimal inference model with two heuristic models, the 'ratio' and the 'difference' model. The ratio model assumes that confidence is a function of the sample proportion alone. This could be the result of a simpler approach in which the population estimate is a point estimate corresponding to the sample proportion which is a more suitable approach in the limit of large samples that are representative of their population[16,17]. The difference model estimates confidence

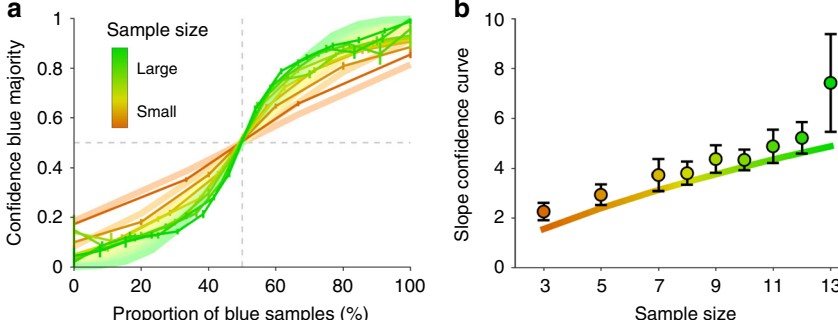

**Fig. 2** Human confidence estimates vary with sample size as predicted by probabilistic inference. **a** Confidence in a blue majority increases with the proportion of blue samples (solid lines), and it does so more steeply the larger the sample size is (color coded). Optimal model is represented in light colors. **b** The slope of the confidence curve in **a** increases with sample size. Participants feature a quantitatively similar increase as the optimal model (solid line). Error bars indicate SEM across participants.

based on the difference of blue and red samples, $N_B$–$N_R$. As the ratio heuristic, this statistic is informative of decision correctness but additionally covaries with sample size, as the ideal observer model, but not directly through sample size. It would correspond to the optimal model if the true proportion would only take two possible values (e.g., 60% blue passengers or 40% blue passengers), i.e. if subjects would discard the variability of the true proportion within each of the two categories (see 'Analytical approximation for Experiment 1' in Supplementary Methods). To account for possible distortions on the response and/or calibrations of heuristics estimators, all model estimates (either from the optimal or heuristics models) were passed through a logistic function that mapped the estimate onto the unity interval. The logistic response mapping was fitted for each model and participant individually (Methods).

The comparison between the optimal model and the ratio model shows that the latter is clearly rejected because of its incapacity to take sample size into account (Supplementary Fig. 4). Even though the confidence estimates of the difference model are sensitive to sample size, they typically do not correspond to the notion of uncertainty that our participants report: the difference model predicts a linear relationship between the sample size and the slope of the confidence curve, while subjects displayed a clear sublinear relationship (see Supplementary Fig. 3). We can thus dissociate the experimental reports from these simple but covariant heuristics and conclude that the response patterns of our participants suggest a probabilistic inference approach. Moreover, as the difference model (corresponding to the optimal response when the variability of true proportion is discarded) can be ruled out, our results suggest that our participants' inference process incorporated not only uncertainty about the passenger majority on the plane (blue or red) but also about its magnitude (the proportion).

**Reliability-based hierarchical integration of ideal observer.** In Experiment 2, participants were told that several airplanes with unknown passenger proportions would arrive at an airport, as before, but that consecutive airplanes would feature correlated passenger proportions because of an event in the city that attracts more travelers of one type. Thus, if the sample of a previous airplane is highly suggestive of a blue airplane proportion, then the participant could not only infer that this previous airplane carries a blue majority, but also that the next airplane is more likely to carry a blue majority, even before observing a sample of passengers leaving it. Importantly, in Experiment 2, there was no feedback about decision correctness of each trial's airplane majority, only an overall score after each block (see Methods and Supplementary Fig. 6b). Inference of an ideal observer in our task

should start with inference of the current context (whether there is a tendency to observe passengers from airplanes with blue or red majorities). Next, this contextual information should be integrated with the current sample to report confidence and decide whether the current airplane it is more likely to hold a red or blue majority (Fig. 3). Thus, the generative structure of the observations that were shown to the participants is hierarchical, with a higher-level variable that determines the context for a block of always five trials, which either favors red or blue airplane majorities, and which in turn generates airplane majorities at the lower hierarchical level across the sequence of trials in the block (Fig. 3a). Both hierarchical levels feature hidden variables that are not observable by the participants. From the generated airplane proportions, samples are drawn, which correspond to the actual observations of the participants (Fig. 3b). Note that the generative process is purely top-down, from the context (high-level hidden variable) to airplane passenger proportions (low-level hidden variables) and then to the samples (observables). However, inference by the ideal observer should first run bottom-up from previously observed samples to infer the value of the contextual variable (Fig. 3c; open nodes) and then top-down from this inferred context (bottom open node) to the variable representing the passenger proportion of the current airplane (orange node). For the ideal observer, this can be formulated as message passing between the hidden variables (Methods). It is worth emphasizing that the task is about inferring the passenger majority of the current airplane, at the lower hierarchical level, rather than asking for the context. Note also that, in contrast to change-point detection paradigms[25] subjects were explicitly told that a new context had to be inferred at the beginning of each block.

As with Experiment 1, we studied how an ideal observer would behave under specific manipulations of the reliability of the currently observed sample through its sample size and the reliability of the context as controlled by the sample size of previously observed airplanes. As with the previous experiment, we first point to patterns of behavior that should be indicative of reliability-based probabilistic inference in our hierarchical task.

First, we expect that confidence in blue majority of the current airplane grows with the proportion of blue samples (Fig. 4a), as in the previous task. However, in addition, we also expect that confidence in a blue majority should be higher in blocks whose actual tendency favors blue airplane majorities, which is indeed the pattern that an ideal observer would show (Fig. 4a). This is because, averaged across trials, the ideal observer can infer what the block tendency is, which on average should be aligned to the true block tendency, resulting in a higher confidence in blue majorities.

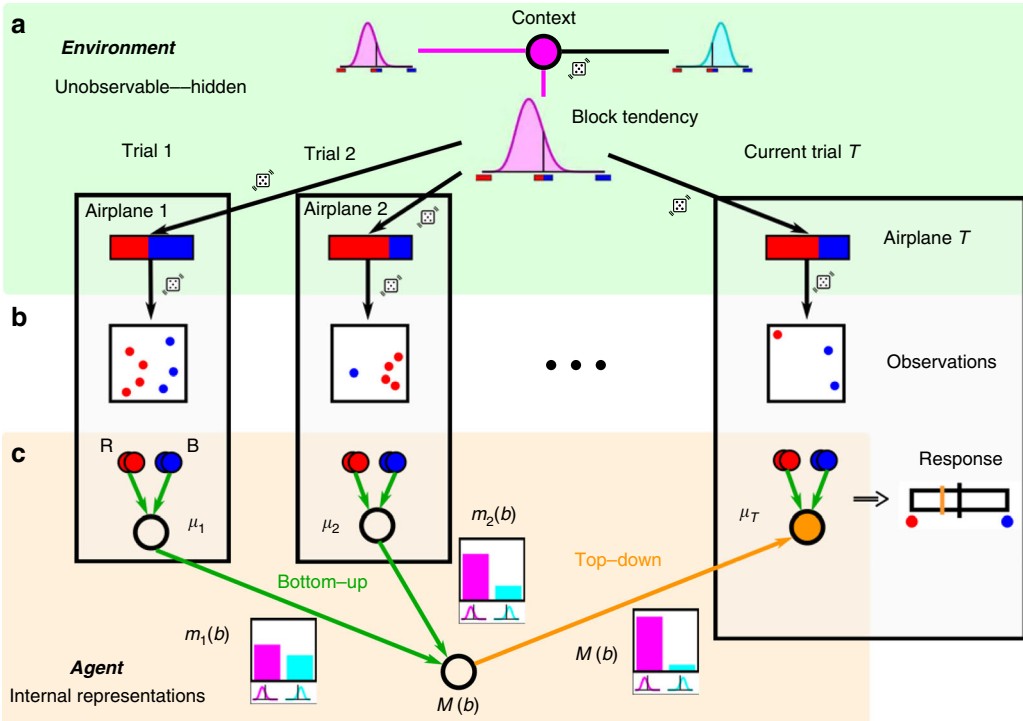

**Fig. 3** Schematic of the hierarchical structure for learning empirical priors Participants are told that across a block of five trials (1, 2,…T ≤ 5) they will see passengers from five different airplanes arriving to the same airport. As before, they are asked to report their decision confidence whether the current airplane carried more red or blue passengers. The schematic illustrates the hypothetical examples of an ideal observer that estimates confidence based on the proportion of blue samples of the current airplane $T$ and on the samples observed in previous trials. The generative model of the observations is as follows. **a** Within a block of five trials, the context, called block tendency, is first selected, which corresponds to choosing either a positively (magenta) or negatively (cyan) skewed distribution over airplane proportions. This context (distribution) is maintained throughout the block of five trials, but on each trial a new blue majority (blue-red horizontal bars indicating the passenger proportion in each airplane) is randomly sampled from that distribution. In the example, the context favors airplanes of red majorities. **b** Sample generation given the airplane majority is the same as for the previous task. **c** The internal representation of the agent (orange background) mirrors the dependence structure in the environment (green background). Probabilistic inference is performed by message passing between the nodes which internally represent the inferred block tendency and the airplane's passenger proportion of each trial (see Methods). Previous trials ($t < T$) provide evidence about the block tendency through the messages $m_t(b)$. They are probabilistically integrated into an overall belief about the block tendency $M(b)$ which provides top-down constraints on the inference of a new airplane's blue proportion (orange node). The confidence in a blue majority of the current airplane $T$ held by the ideal observer (response bar, right) should follow from both the current sample proportion and the inferred block tendency from previous samples.

Second, averaged across sample proportions and samples sizes, confidence in a blue (red) majority in the current airplane should increase the higher the inferred tendency of blue (red) passengers is. Because of the symmetry across these two cases, we defined a (block-) aligned confidence to indicate the confidence in the direction (passenger type) that is aligned to the actual block tendency and pooled the results across these two cases. For the ideal observer, aligned confidence increases with the aligned inferred tendency (Fig. 4b). In other words, the inferred context informs inference of the current airplane's proportion to the degree that the context is reliable itself.

Sample size of the current observation should play a very important role in modulating decision confidence as it indicates increased reliability of the sample relative to the prior. Indeed, aligned confidence increases with the aligned sample proportion, and it does so with a higher slope when sample size is large (Fig. 4c). Similarly, if the context is inferred probabilistically, the reliability of previous trials should be taken into account. As a consequence, the sample size of the previous observation should modulate aligned confidence (Fig. 4d). For instance, if the previous sample was large and suggested a red majority, then confidence in a red majority in the current trial should be larger.

Another pattern that is expected from the ideal observer is that the weights (see Methods) of all previous trials in a block onto the confidence in the current trial should be constant (Fig. 4e), because an earlier trial provides the same evidence for the context as a recent one, on average across blocks. Finally, the more trials have been observed in the block, the better the inference about the current context ought to be. Thus, on average across blocks, aligned confidence should increase with the number of previous observations which indicates accumulation of evidence for the contextual variable (Fig. 4f).

It is important to emphasize that these patterns correspond to predictions of the ideal observer model. They will be used as a benchmark for a direct comparison to behavioral data without fitting any parameters. Consequently, we do not expect a perfect match in terms of absolute values, but we would expect similar patterns of variation if participants follow a probabilistic inference strategy.

**Human behavior follows patterns of probabilistic inference.** We first tested whether human participants can infer and use contextual reliability by studying whether they followed the patterns described above. We found that our participants' confidence in a blue majority increased with the proportion of blue samples, but that confidence in a blue majority was larger when the block favored airplanes with blue majorities as opposed to red majorities (Fig. 5a). This result indicates that participants not only

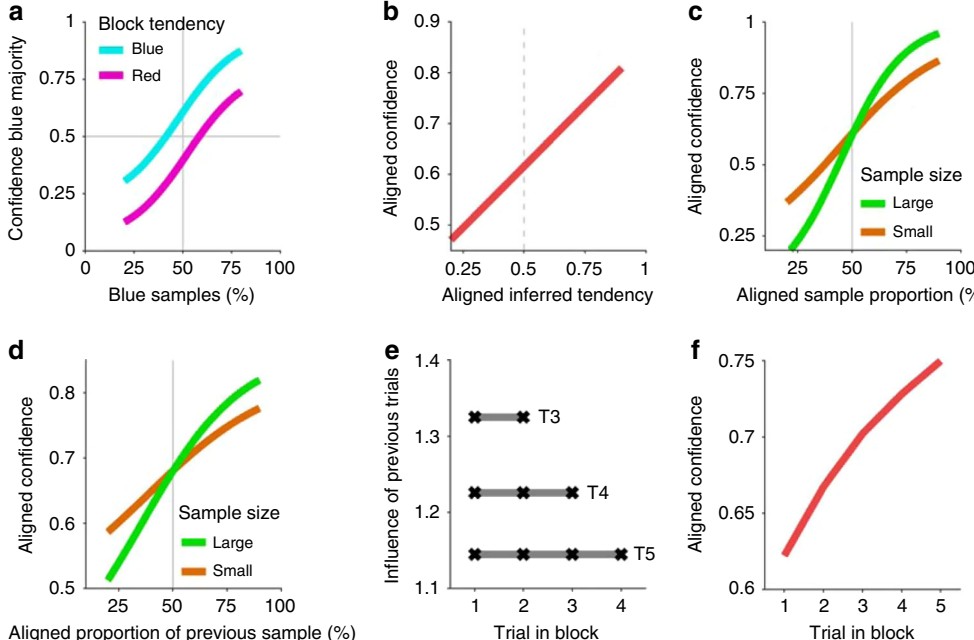

**Fig. 4** Characteristic behavioral patterns of probabilistic inference in the hierarchical inference task **a** Confidence in a blue majority of the current airplane (current trial) should increase with the proportion of blue samples, as in the previous task, but in addition confidence should be larger in a block that favors blue majorities (cyan) than in a block favoring red majorities (magenta). **b** Information of the block tendency should gradually increase the confidence in the corresponding trial majority. Thus, responses can be pooled with respect to the real block tendency. We refer to it as 'aligned confidence' and use the same concept for other relative quantities below. **c** Confidence in the aligned airplane majority increases with the aligned sample proportion. This modulation is stronger for larger sample sizes (green) compared to smaller ones (orange) while it has no effect for an indifferent sample (50% sample proportion, crossing point between the two lines). **d** Likewise, aligned confidence increases with the aligned sample proportion of the preceding trial and is modulated by its respective sample size. **e** The influence of all previous trials, determined by the weights of a regression analysis, should be equal on average (e.g., trials 1–2 on trial 3, T3). However, it decreases with the number of previous trials due to normalization. **f** Aligned confidence increases across trials within a block because evidence for the block tendency accumulates across trials in the block. All patterns are derived from the ideal observer model (see Methods).

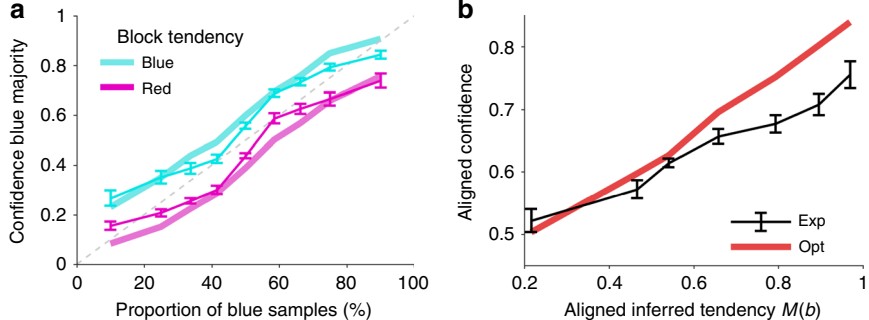

**Fig. 5** Inferred block tendency affects confidence reports. **a** Confidence in blue majority is higher when the block tendency favors blue majorities (cyan) than when it favors red majorities (magenta). Experimental results (data points) are shown along with optimal behavior (solid lines), indicating an integration of sample information with a learned belief about the block tendency. **b** Aligned confidence (black) increases with the optimally inferred belief about the block tendency and is a close correlate of the optimal response (red), suggesting that participants internally track a graded belief based on previously available evidence. Error bars indicate SEM across participants.

relied on the current sample to infer the current airplane majority, but that they also inferred the context and used it to modulate their confidence judgments.

Further evidence for this result comes from the observation that aligned confidence increases with the strength of the inferred tendency aligned to the block as computed by the ideal observer, indicating that the more evidence was collected for a given block's tendency, the larger the modulation on the confidence reported in the current trial was. The gradual increase (which was also present at an individual level, Supplementary Fig. 8a) shows how nuanced the representation of the contextual variable is as there is

no thresholding nor any sign of categorical representation. This shows that the contextual variable—for which we never explicitly asked—is represented in a graded manner, as it would be expected from a probabilistic agent. Our participants not only followed this pattern qualitatively, but they also seemed to adhere quite closely to the quantitative, parameter-free, predictions made by an ideal observer (Fig. 5b; Pearson correlation on binned values, pooled across participants, $\rho = 0.77$, $p = 5.13 \times 10^{-33}$), except for the fact that contextual information did not affect predicted confidence as much as when contextual information was high (Fig. 5b, rightmost part), which was also observed on a

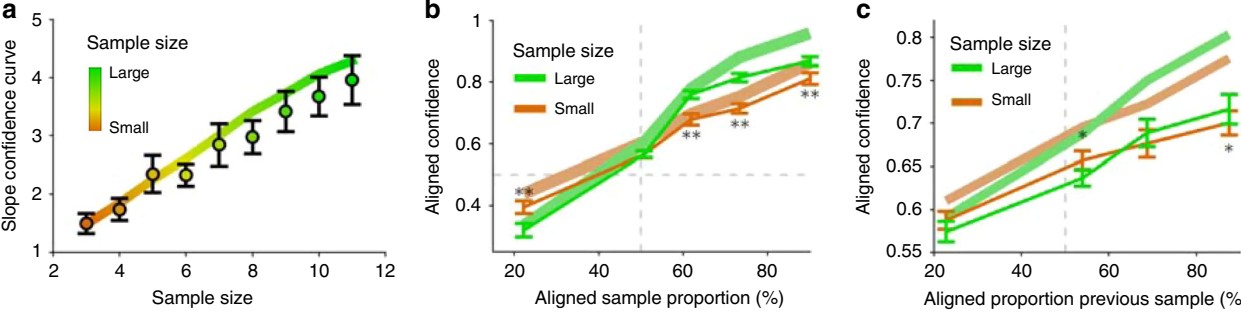

**Fig. 6 Sample size effects reveal reliability-based information integration. a** As in the basic task (Fig. 2b), the slope (data points) of the confidence curves over the sample proportion increases with sample size and tightly follows the optimal pattern (solid line). **b** The modulation of aligned confidence with the aligned sample proportion of the current trial is larger when the sample size is high (green) than when it is low (orange). Significant signed differences of a bin-wise one-sided signed rank test are indicated, *$0.01 < p \leq 0.05$, **$p \leq 0.01$. **c** The modulation of aligned confidence with the aligned sample proportion of the previous trial is larger when the sample size of the previous trial is high (green) than when it is low (orange), similar to the previous panel. Error bars indicate SEM across participants in **a–c**.

participant by participant basis (one-sided signed rank on fitted slopes, $p = 0.004$). Thus, even though the inferred tendency is subjective to the participant, the correlation with the inferred tendency of the ideal observer shows that participants must be tracking a similar quantity.

Next, we studied how reliability governs hierarchical information integration (see Fig. 4c, d). Both the current sample and previous samples should be relied upon more strongly when their reliabilities, controlled by sample size, are high. We first confirmed that the slope of the confidence curve increases with sample size of the current observation (Fig. 6a; Pearson correlation of slope with sample size, pooled across participants, $\rho = 0.49$, $p = 8.67 \times 10^{-14}$), indicating that participants used the reliability of the current observation to form confidence estimates, as in the previous task without hierarchical dependencies (see Fig. 2b).

Beyond the finding above that participants learn the block tendency (Fig. 5a), they should use it selectively and rely more strongly on the sample compared to prior information when sample evidence is reliable (Fig. 6b, pattern: Fig. 4c). Indeed, the modulation with the aligned sample proportion is stronger for larger sample sizes and leads to the crossover of the two conditional curves (signed difference of conditional slopes from linear regression, signed rank test across participants, $p = 1.44 \times 10^{-5}$). On average across trials, prior information increases aligned confidence (Fig. 6b). Relative to this offset, behavior is less strongly driven by smaller samples because they provide less information so that the agent resorts more closely to the top–down expectations gained from previous trials.

Direct control of the reliability through sample size allows us to study whether the inferred reliability of the context interacts with the reliability of the current observation to inform confidence judgments. Using this degree of freedom, we tested whether participants used the reliability of the previously observed sample. We found that, consistent with the pattern predicted by the ideal observer, aligned confidence increased with the aligned sample proportion of the previous sample and that this increase was larger the larger its corresponding sample size was (Fig. 6c; signed-rank test for positive difference of linear regression slopes across participants, $p = 0.002$; see also dependence on previous message $m_{t-1}(b)$ Supplementary Fig. 8b).

A central prediction of the probabilistic model is that all previous trials should have equal influence on behavior on average across blocks (see Fig. 4e, f). We determined their influence from a regression analysis on the confidence judgments (see Methods) and found a rather balanced influence of all previous trials (Fig. 7a). Accordingly, no significant trend could

be evidenced through another linear regression analysis in which the previous trial index is used to predict the average weight of the previous trial on the aligned confidence (regression on the means across participants, separately for current trials position 3, 4, and 5: $p$-values 0.41–0.89 for trials with 2–4 previous trials respectively). Apparently, there are no signatures of temporally selective evidence integration for the contextual variable such as a confirmatory bias, which is characterized by an insufficient belief revision once a belief has been established. If it were present, later trials would be expected to have a lower influence here. Probabilistic inference on the other hand, never fully collapses onto one specific interpretation and hence never excludes evidence for competing hypotheses. Similarly, this rather balanced weighting is also inconsistent with some sort of leaky prior integration scheme in which evidence presented long ago is fading from memory. In agreement with these findings, evidence for the block tendency, and thus also aligned confidence, increases over the trials within a block (Fig. 7b). A linear regression analysis of aligned confidence as a function of the aligned trial index clearly shows the expected increase (regression on means across participants, $p = 8.68 \times 10^{-9}$). Overall, hierarchical integration offers a parsimonious explanation for context integration which does not require explicit memorization of previous samples after they are integrated into the context-level variable.

Interestingly, the most obvious quantitative departure from the expected patterns was that human participants appear to rely less on contextual information as the observed effects of previous trials were smaller than the predictions from the ideal observer. For instance, the effect of previous trials on aligned confidence is weaker (see e.g., Fig. 7a) but does not depend on how long ago the information was acquired. Further support for such an insensitivity to prior information is provided by trials in which an ideal observer would e.g., estimate a red majority despite more blue samples because of a high prior belief in a red tendency. We found that most participants make these evidence-opposing choices (see Methods, one-sided signed rank test with respect to non-hierarchical ratio model with realistic response noise, $p = 0.007$; Supplementary Fig. 5b). There is however a tendency to stay on the side of the category boundary that is suggested by the momentary evidence, as they make significantly fewer opposing choices than the optimal model (one-sided signed rank test, $p = 0.008$).

Finally, we tested whether this relative insensitivity to prior information could be explained by mismatching assumptions about the magnitude of the block tendency which we modeled with specific skewed distributions of passenger proportions under

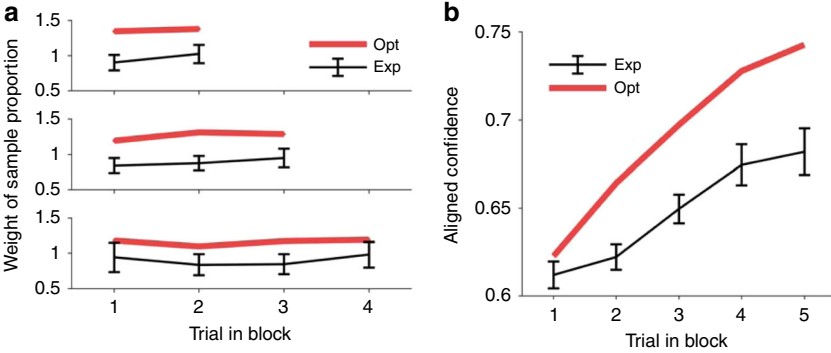

**Fig. 7** Behavior reflects hierarchical evidence integration across trials. **a** On average across blocks, all previous trials provide the same information about the block tendency irrespective of their temporal distance to the current trial. From top to bottom, trials number 3–5 of each block are predicted from the indicated previous trials (sample proportion). Participants show a balanced weighting despite smaller weights compared to the ideal observer model (red). **b** Participants accumulate evidence about the block tendency in a gradual fashion. Aligned confidence increases over trials within a block despite a smaller effect compared to the optimal model (red). Error bars indicate SEM across participants.

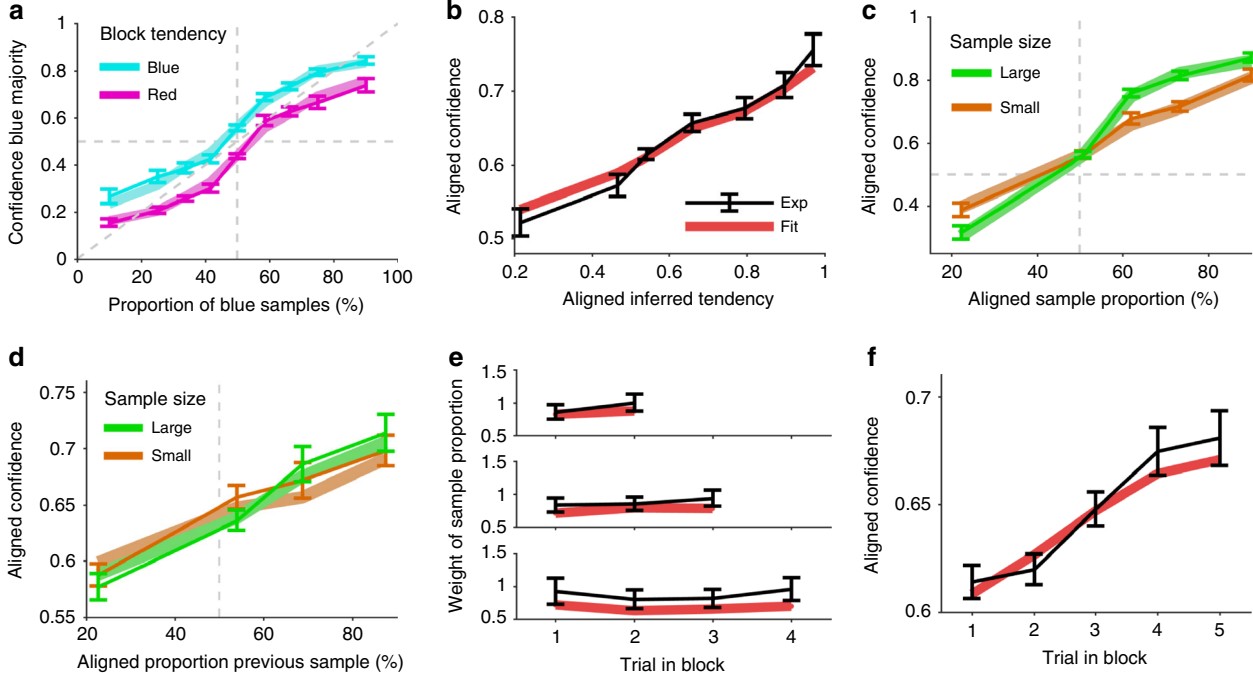

**Fig. 8** Patterns for fitted block tendency. Behavioral patterns in the hierarchical inference task (Experiment 2) compared to a fitted model assuming a mismatched block tendency and a sigmoidal response mapping accounting for distortions on the response. The fits of this model closely reproduce the patterns produced by participants. Error bars indicate SEM across participants.

the red and blue contexts (see Fig. 3a). In fact, some behavioral biases, such as confidence under- and overestimation[26], can be partly explained by choosing (structurally) mismatched probability distributions for the task at hand[27,28]. To test this possibility, we used a model that allowed for a differently skewed distribution implementing this block tendency (see Methods) and compared it to the ideal observer model. To correct for other distortions, both models used an additional mapping onto the final response. We found that the model with the mismatched block tendency almost perfectly described the patterns of probabilistic inference (Fig. 8; exceedance probability $p \approx 1$, for patterns see Supplementary Fig. 7) and that participants appear to subjectively assume a weaker block tendency as evidenced by the expectation value of the skewed Beta-distribution used to model a blue block tendency (optimal 0.61, median across participants 0.55, one-sided signed rank test for difference, $p = 1.68 \times 10^{-4}$). This suggests that qualitative differences arise from a mismatch

between the experimental and the assumed skewed distributions by the participants.

**Model comparison favors probabilistic inference of context.** The previous analysis has shown that behavior adheres to the main features of probabilistic inference in a reliability-based hierarchical task. We have seen that these patterns were qualitatively reproduced by the optimal model without the fitting of free parameters, and that a simple extension of the ideal observer model largely improved the qualitative fits of the patterns.

To go beyond qualitative patterns of behavior and provide a more quantitative account of the results and the adherence of behavior to reliability-based hierarchical inference, we fitted the ideal observer estimate of the contextual variable to behavior and compared it to simpler heuristic estimates that do not rely on probabilistic inference. These simpler models assumed specific forms for the accumulated contextual information that depart

from the optimal computations, as follows (contextual variable $M = M(b)$ in Fig. 3; see Methods).

In the 'averaging' model we assume that the estimate of the contextual variable $M$ equals the presented percentages of previous trials in a block and thus neglects sample size. In the 'tally' model we assume that the estimate of the contextual variable $M$ equals the ratio between the total number of blue samples observed so far in all previous trials within a block over the number of all red and blue samples observed within a block so far. This is similar to pooling the samples of all trials, as if they were drawn from a common population. Thus, as larger samples contribute more points, this model is sensitive to sample size, but in a different way than the ideal observer model. Finally, in the 'difference' model, contextual information is a sigmoidal function of the running average of the differences between the number of blue and red samples in all previous trials. All these models only differ in how they estimate the contextual variable $M$. To introduce as few constraints as possible on the integration of $M$ with the current sample $(N_B/N, N)$ and to compute the final response, we used a flexible generalization Eq. (14) of the sigmoidal response mapping Eq. (13), attempting to reduce noise for model comparison. Even though all three heuristic approaches are close correlates of the optimally estimated contextual variable, we found that the three models were inferior to the probabilistic strategy of the ideal observer model (Supplementary Fig. 3).

Beyond quantitative comparisons, heuristic models also failed to quantitatively reproduce the defining features of subject behavior. Specifically, participants' responses were influenced by the proportion of blue passengers in previous trials of the block, and that influence increased with trial position in the block as subjects accumulated evidence about current context across trials (Supplementary Fig. 10). Such feature was seen in the optimal model. By contrast, in all three heuristics models, the influence of the proportion of blue passengers in previous trials remained constant across the block, as in these heuristics models evidence about the current context is averaged and not accumulated across trials (see Eqs. 9–11; Supplementary Fig. 10c-e). Moreover, the 'averaging' model was, by construction, insensitive to the sample size of previous trials, unlike our participants (Supplementary Fig. 10b).

## Discussion

One important question is whether humans can hold probabilistic representations of contextual variables and use them to improve inference of lower-level variables by providing suitable constraints on their possible values. Here, we report that humans can perform reliability-based hierarchical inference in a task in which they have to report their decision confidence about the value of a lower-level variable that is constrained by a higher-level, partially observable variable. We controlled evidence by using reliability cues in the form of sample size, giving us enough leverage to test the identified patterns of hierarchical probabilistic inference. The similarity between observed and probabilistic inference patterns of behavior, the strong dependence of confidence on currently and previously observed samples sizes, and a model comparison between optimal and heuristic models, supports the notion that humans can mentally hold and update ubiquitous representations of uncertainty over structured knowledge representations such as graphs[22].

A large body of research has addressed the question whether, and under what conditions, humans can perform probabilistic inference, typically, by using perceptual tasks[10,29,30]. More recently, the usage of confidence reports has opened a window to more directly examine how uncertainty is handled in internal

models that humans use while they perform a task[8,23,28,31,32]. However, most of this work has focused on simple inference problems in which the value of a hidden variable has to be estimated based on noisy evidence[24,33,34], without any hierarchical structure. In contrast, even visual processing in normal conditions should rely on hierarchical schemes where hidden variables at higher levels constrain the values of partially observed variables at lower levels[35]. Hierarchical representations allow to exploit inferential constraints by learning them from experience with related situations by exploiting abstract similarities through contextual variables. Such joint inference over structured probability distributions is a crucial ingredient for theories such as predictive coding[3,6,36]. However, whether human inferences rely on ubiquitous probabilistic representations across a hierarchy of variables is largely unknown.

Addressing this important question requires the ability to independently control the reliability of higher-level and lower-level variables to test, for instance, whether and how behaviorally reported confidence is modulated by them. If reliability cues produce modulations of confidence reports in accordance with theoretically predicted patterns, then such observations would constitute evidence in favor of mental representations similar to probabilistic graphical models. Previous work has studied perception and decision making in similar hierarchical schemes like ours[1,7,8,15], but it has been difficult to independently modulate the reliability at both higher and lower hierarchical levels. For example, when using stimulus duration and stimulus strength as an indirect proxy to control reliability[15], the way these manipulations affect reliability depends on the specifics of the sensory system and sensory noise. In our task, uncertainty emerged not from sensory noise but from a hidden cause for stochastically generated stimuli, and the reliability of both levels could be controlled directly and independently through sample size, thus providing an objective measure of trial-to-trial reliability independent of the sensory system. Our task revealed that humans modulate their confidence not only based on the reliability of the currently observed sample, but also on the inferred reliability of the context which is itself a function of previous samples. Specifically, we have found strong dependencies of confidence on the sample size of current and previous observations, and these dependencies adhered to the predicted trends and patterns of hierarchical probabilistic inference. Dependencies on previous observations emerged only based on the distribution of previous stimuli, without any trial-to-trial external feedback that could be used to modulate the priors. In summary, while previous studies had already shown in isolation sample size sensitivity[37] and some form of hierarchical probabilistic reasoning[1,7,8,15], here the conjunction of both phenomena and the very detailed correspondence between human and optimal behaviors builds strong evidence for ubiquitous reliability-based integration of hierarchical information, even without extensive prior training on the task.

It is possible that our participants did not truly hold probabilistic uncertainty representations over a mental graphical representation across multiple levels, but that they rather used very sophisticated heuristics that we were not able to characterize. However, estimating uncertainty about latent variables is a particularly difficult problem for heuristic approaches just based on point estimates that disregard the distributional format that the estimate should take[5], e.g., that several airplane proportions are consistent with a given sample. In our task, for instance, learning calibrated confidence reports would require repeated exposure to the same sample together with supervising feedback about the actual latent variable (airplane majority). Even for very simple problems, the scarcity of such data makes this frequentist approach to uncertainty estimation practically difficult and thus

un-ecological. As we did not provide supervising feedback, our participants presumably held accurate internal trial-by-trial representations of uncertainty[38,39]. Although we cannot completely rule out the use of non-probabilistic or heuristic shortcuts, the main patterns of probabilistic inference have been fulfilled by our participants. Their generalizations are hard to conceive without relying on an internal generative model of the observations. This is in line with previous studies (e.g.,[40,41] which conclude that human inferences are model-based or use internal simulations[42].

One clear limitation of our study is that it shows that humans can use reliability-based hierarchical integration of evidence, but it does not speak to the circumstances when this occurs. In particular, our results contrast with a vast literature that has reported deviations from the norms of rational inference in human judgments such as sample size insensitivity[17,19,43]. One important methodological difference between this previous work and ours is that behavioral economics has typically dealt with situations that have been conveyed using mathematical terms[21]. We speculate that the success of our participants in 'understanding' the hierarchical structure of the task is the result of the way the task has been framed and communicated. We put participants in an imaginary yet intuitive setting of arrivals to an airport whose city hosts an event and refrained from using terms such as "urns" or "correlations", which mathematically define our task on an abstract level. Evidently, this approach was successful in at least two respects. First, the task structure is clearly communicated so that participants make roughly correct assumptions for inference. Second, our participants managed to interrogate cognitive systems that are capable of probabilistic inference[14]. Interestingly, a recent proposal has suggested that intuitive tasks that sidestep high demands on working memory and natural language may improve performance[44]. The existence of such framing effects[45] onto the algorithmic nature of perceptual inference mechanisms should be tested in a future experiment. A related but slightly different hypothesis is that probabilistic inference would be shaped during lifetime experience by repeated exposure to choices between options that require integration between sources of varying reliability. In the lab, such probabilistic inference process would only be applied if the task bears some similarity with the problems already encountered by the subjects in their life. In support of such hypothesis, a recent study did find sample size sensitivity in how subjects updated product evaluation by learning about previous consumers' ratings[37], which is a highly familiar task routinely performed in everyone's everyday life.

However, our work has also revealed some differences between optimal and observed behavior. Most strikingly, we have found evidence that top-down information is relied upon less strongly relative to information from the specific instances of the sample[28]. Such a tendency to discount prior information is indeed reminiscent of the biases that emerge when the representativeness heuristic is used[16,17]. However, as we have shown with a model that assumes a different block tendency (Fig. 8), all these differences could be attributed to mismatched assumptions about the prior distribution of the context. In general, when comparing behavior against normative approaches, the interpretation of deviations should consider as much as possible the internal assumptions, constraints and motivations that the participant obeys[46,47]. Accounting for such differences might be crucial to interpret and possibly account for many cognitive biases[27,48]. Beyond these differences in the central inference stage, there could be alternative sources of distortions in the conversion from the estimate into a motor report. Taking into account such distortions allowed to capture some other part of the departure of our participants' behavior to the optimal observer

(Supplementary Fig. 1). By contrast, our participants behavior was found to be little affected by numerosity or other forms of sensory noise (see 'Sensory noise' in Supplementary Methods).

The easiness with which our participants seemed to perform probabilistic inference over the mental representation of a graphical model at several levels of a hierarchy should not distract us from the computational difficulty of the inference process. Typically, probabilistic inference even in simpler tasks involves complex operations such as normalization and marginalization[5,49,50]. Interestingly, inference in our task can be considerably facilitated if the conditional independence properties between variables are exploited. In this case, the distribution factorizes so that only local computations (marginalization) need to be performed whose results can be passed on as messages. Hence, the graphical structure of the model facilitates inference which may even be implemented with recurrent neural populations[51].

Apart from the tractability of the computations, we must bear in mind that the goal of the participant is not necessarily pure inference, but the maximization of some subjective cost-benefit measure[52]. Further research is needed to test what constitutes the main challenges to probabilistic inference for humans such as imposing adequate structural constraints that leverage contextual knowledge or the use of tractable approximations due to limited cognitive resources.

In sum, we have developed a novel reliability-based hierarchical task based on which we found that humans are sensitive to the reliability of both high- and low-level variables. Our results reveal uncertainty-sensitive integration of information in hierarchical state representations and suggest that humans can hold mental representations similar to probabilistic graphical models where top-down and bottom-up messages can inform behaviorally relevant variables.

## Methods

**Participants**. All participants were invited to complete three sessions on different days within three consecutive weeks. The sessions were targeted to take about 35 min (Session 1) and 45 min (Sessions 2,3). In total 25 participants (15 female, 10 male) were recruited mainly among students from the Pompeu Fabra University in Barcelona. The study was approved by the Ethics Committee of the Department (CIREP approval #0031). We excluded data from one participant that did not complete the experiment. The median age was 25 (minimum 20, maximum 43). We accepted all healthy adults with normal or corrected to normal vision. We obtained written confirmation of informed consent to the conditions and the payment modalities of the task. Irrespective of their performance, they were paid 5 € for session 1 and 7 € for sessions 2 and 3.

Additionally, they had the chance to obtain a bonus payment which was determined by the mean of their final score after removing the worst trials (2.3%). The score $S = 1 - |y - y_{opt}|$ of a response $y$ was computed based on the proximity to the optimal confidence report $y_{opt}$ (see below for details of the optimal model in both experiments). As such, the overall score reflected the ability of the subject to correctly infer the probability that the observed stimuli would be sampled from one category or the other. The payment was determined by comparison to an array of five thresholds that were set according to the {0.5, 0.6, 0.7, 0.8, 0.9} cumulative quantiles of the empirical score distribution across prior participants. A higher score $S$ corresponds to a better performance so that participants were paid an additional bonus of {1, 2, 3, 4, 5} € if their final score was higher or equal to the quantile thresholds. This is a way of rewarding their efforts to optimize their responses.

Written task instruction explained that we would score their responses with respect to the chances that their decision would be correct and that bonus payments would be based on that score. Additionally, they were informed that their score was to be compared to the other participants and that the experimenter could monitor their behavior on-line via a second screen from outside.

**Stimuli & responses**. The task was presented on an LCD screen with a computer running Matlab Psychtoolbox 3.0.12. Immediately after trial onset, our participants were shown the sample consisting of red and blue solid circles arranged on a two-dimensional grid about the screen center (Fig. 1a). The only feature that distinguished the sampled passengers was the dot color that we chose to be either blue or red. Because the positions of the dots are communicated not to be informative, the sample is completely summarized by the sufficient statistics. We tried to make

the number of dots (sufficient statistics in our task) easily perceptible while making their locations appear as random as possible. Adequate grid spacing was introduced to prevent the circles from overlapping. Furthermore, we kept red and blue samples separate along the horizontal direction (details in SI).

The display is static until the participant makes a response by clicking the USB-mouse which clears the display of the sample. After a short delay of 300 ms, the program shows a centered horizontally elongated response bar of random horizontal extent with a vertical line marking its center. In addition, the response cursor (Fig. 1a, orange vertical line) is shown at a random and uniformly distributed initial horizontal position along the response bar. Participants can adjust the horizontal position of the response cursor by moving the mouse horizontally and confirm the input with a click to report their choice about the airplane's passenger majority and their subjective confidence in its correctness. The movement range of the response cursor was bounded to the horizontal extent of the response bar. The raw response is linearly mapped onto an interval between [0,1] and interpreted as the confidence in a blue trial majority $y$. Consequently, the corresponding quantity for the confidence in a red majority is $1 - y$.

**Experiment 1: Procedure & instructions**. First, participants read detailed written instructions of the task. We introduced the task metaphor that relates to judging the (hidden) majority of passengers on a flight and used it to explain the mathematical assumptions in more intuitive terms (see Supplementary Methods).

Additionally, our participants were given 30 trials to familiarize with the handling of the task. The subsequent experimental session (session 1) consisted of 280 trials with pauses together with feedback after every 5 trials. The sample sizes $N$ were independent and identically distributed (i.i.d.) samples from $\{3, 5, 7, \ldots, 13\}$ while the hidden airplanes' passenger proportions $\mu$ were i.i.d. samples from a Beta (4,4) distribution. Then, the number of blue passengers of the sample is determined by a draw from a Binomial distribution $N_B \sim \text{Bin}(N, \mu)$. After each trial, the participant receives feedback about the correctness of his decision (whether the cursor was placed on the side corresponding to the underlying passenger majority) but no supervising feedback regarding his confidence estimate. In addition, a two second time-out was presented for incorrect decisions which is signaled by a horizontal 'progress bar' which linearly diminishes over time indicating the fraction of the waiting time left. During time-out, there is nothing a participant can do to proceed but wait. In principle, the correctness feedback could be used by participants to learn the mapping from stimuli to the probability of selecting the correct category. In practice however, subject behavior was found to be very stable from the first test trial and throughout the session (Supplementary Fig. 6).

Every five trials, a pause screen was shown which provided information about how many out of all trials had already been completed. To motivate engagement in the task, we gave motivational feedback as an average $\langle S \rangle$ of the score $S$ (distance to optimal observer, see above) over the last 5 trials since the last pause. Such feedback was uninformative as to how subjects should change their behavior to improve their score: Because it averaged performance over 5 trials, it was very unlikely they could use to learn current mappings and shape future responses (see stability of participants behavior Supplementary Fig. 6). Additionally, they also received a time-out of a few seconds proportional to $1 - \langle S \rangle$. The overall rationale behind the time-out was to more strongly incentivize task engagement and prevent click-through.

**Experiment 2: Generative model for the stimuli**. In Experiment 2, trials of one block are tied together because they depend on a common unobserved variable selecting the context. There were two possible contexts: one biased towards red passengers, the other towards blue passengers. To keep the notation simple below, we use the same variable names for the generative process (Fig. 3a) as for the ideal observer (Fig. 3c), although in general, an agent's representation is not necessarily the same as the generative process in the environment. First and once for every block, the binary variable $b$ governing the prevalence for either red ($b = 0$) or blue ($b = 1$) passenger majorities in the airplanes, called block tendency, is drawn from a Bernoulli distribution $b \sim \text{Bernoulli}(0.5)$. Then for every trial, the unobserved proportion of blue passengers of the airplane $\mu$ is drawn from a mixture of two Beta distributions depending on the block tendency $b$.

$$p(\mu|\nu_1, \nu_2, b) = b \cdot Beta(\mu|\nu_1, \nu_2) + (1 - b) \cdot Beta(\mu|\nu_2, \nu_1). \quad (1)$$

The Beta distribution is parameterized by two parameters ($\nu_1 = 14, \nu_2 = 9$), chosen such that the resulting distribution over the passenger proportion $\mu$ is skewed. By convention, $Beta(\mu_t, \nu_1, \nu_2)$ is negatively skewed ($\nu_1 \geq \nu_2$) and models a blue block tendency. The greater the expectation $\nu_1/(\nu_1 + \nu_2) \approx 0.609$ the more extreme this effect because more airplanes with a majority of blue passengers ($\mu > 0.5$) as opposed to red passengers ($\mu < 0.5$) will be encountered.

Once the block tendency $b$ has been selected in a block, sampling of the observed passengers in the following 5 trials within a block proceeded as in Experiment 1. First, the sample size $N$ is determined by an i.i.d. drawn from a uniform categorical distribution $Cat(N|1/n, \ldots, 1/n)$ over all $n$ sample sizes $N \in \{3, \ldots, 11\}$. Then, the number of blue passengers of the sample is determined by a draw from a Binomial distribution $N_B \sim \text{Bin}(N, \mu)$. Hence, the distribution for

each of the 5 trials within a block is

$$p(N_B, N, \mu|\nu_1, \nu_2, b) \propto Bin(N_B|N, \mu) \cdot Cat(N|1/n, \ldots, 1/n) \cdot p(\mu|\nu_1, \nu_2, b). \quad (2)$$

The geometric placement on the screen is not considered to be part of the generative model as we assume that only the sufficient statistics matter. The expression in Eq. (2) defines the probability distribution for the sufficient statistics of the observations of trial $t$ to which we refer more concisely by $p(q_t, N_t, \mu_t|b, \nu_1, \nu_2)$, thus equivalently expressing it in terms of each trial's sample proportion $q = N_B/(N_B + N_R)$ of the number of blue ($N_B$) and red ($N_R$) passengers, and the sample size $N = N_B + N_R$. We drop the conditioning on the parameters of the categorical distribution over sample sizes to keep the notation uncluttered. Using this expression, the entire sampling distribution over all variables of all trials within a block is:

$$p(q_1, \ldots, q_5, N_1, \ldots, N_5, \mu_1, \ldots, \mu_5, b|\nu_1, \nu_2) = p(b) \prod_{t=1}^{5} p(q_t, N_t, \mu_t|b, \nu_1, \nu_2).$$

$$(3)$$

Note that given the block tendency $b$, the per-trial quantities, such as $\mu_t$, are conditionally independent.

**Experiment 2: Procedure & Instructions**. Experiment 2 comprises the sessions 2 and 3 and was carried out with the same 25 participants as in Experiment 1 (session 1). Despite the hierarchical extension across blocks of five trials, the handling of the task and the presentation of the sample is virtually the same. The changes to the latent structure should lead to a different interpretation of the information which we attempted to convey by an extension of the task metaphor (see Supplementary Methods) and written task instructions.

As for Experiment 1 and prior to starting session 2, participants completed two very short training sessions. First, they were given 20 trials (4 blocks) with a strong and visually obvious block tendency (sample sizes $\{8, \ldots, 11\}$, block tendency Beta(15,7)). Then another 30 trials under slightly harder conditions (sample sizes $\{3, \ldots, 11\}$, block tendency Beta(15,7)). Importantly, this only permits them to understand the structure of the reasoning task, such as the dependence between the variables, and get familiarized with the task environment in increasingly difficult conditions. We intentionally provided as little information as possible as to how they should respond. The important point was to make clear what the structure of the process was that generated the samples. Thus, we did not monitor their performance, nor give them any feedback about how specifically they should place the response cursor. They could, however, ask the experimenter to clarify the assumptions behind the task. We proceeded to the actual experimental session when our participants reported that they had 'understood' the task. The above-mentioned procedure was clear enough to achieve that, and yet sparse enough not to reveal the normative response strategy against which we wanted to compare their behavior.

After familiarizing, our participants completed 270 trials of the experimental session 2 with an even more difficult setting of the parameters (sample sizes $\{3, \ldots, 11\}$, block tendency Beta(14,9)). On the third session, on a different appointment, the participants just continued the instructed task of session 2 for 300 trials with identical settings to obtain more data.

In Experiment 2 and different from Experiment 1, no feedback nor time-out was provided after each trial. However, as in Experiment 1, every five trials, i.e., after each block in Experiment 2, participants were presented with a pause screen with a score based on the results of the last block and a time-out of a few seconds proportional to $1 - \langle S \rangle$. As described before, the purpose was mainly to engage participants with the task. That they may have used this extremely sparse and indirect information to somehow guide future responses in Experiment 2, seems even more unlikely than in Experiment 1 as participants already showed no signs of converging to the normative strategy over time there (Supplementary Fig. 6b) where the task was less complex than in Experiment 2 with several hidden variables.

**Ideal observer for Experiment 1**. The ideal observer model is assumed to know the actual generative process of the observations. Based on the observed passengers, it infers the most likely airplane proportion. Due to the choice of a conjugate prior distribution $p(\mu)$ for the Binomial probabilistic model $N_B \sim \text{Bin}(N, \mu)$ above, posterior inference yields a Beta-distribution over the latent airplane proportion $\mu$. Specifically, to give calibrated responses, i.e., confidence estimates that correspond to the actual odds of making correct decisions, the prior distribution used for inference must correspond to the actual base rates specified by $Beta(\mu|4, 4)$. The confidence in e.g., a blue trial majority $c(B)$ of an ideal observer can be expressed as the belief that choosing a blue majority is correct by integrating over the corresponding subspace[23] of inferred blue majorities:

$$c(B) = 1 - c(R) = p(\mu > 0.5|N_B, N_R) = \int_{0.5}^{1} \text{Beta}(\mu|N_B + 4, N_R + 4)d\mu \quad (4)$$

**Heuristic models for experiment 1**. Here we describe two heuristic models that humans could use to estimate the probability of blue passenger majority on the airplane.

1.  Ratio model
    In the ratio model, the response is simply mapped from the proportion of blue passengers in the sample $N_B/(N_B + N_R)$

    $$c(B) = \sigma(2N_B/(N_B + N_R) - 1)$$

    where $\sigma$ is a sigmoid function with possible distortions that provides output in the [0 1] range (see below).
2.  Difference model

    In the ratio model, the response is mapped from the difference between the number of blue and red passengers in the sample $N_B - N_R$. Again the difference is mapped onto the [0 1] range using a sigmoid with possible distortions:

    $$c(B) = \sigma((N_B - N_R))$$

**Distorted reports of internal confidence estimates.** Apart from inference, behavior may be influenced by extraneous factors, e.g., due to motor control constraints. We accounted for those by a nonlinear transformation of the confidence estimate $c \in [0, 1]$ onto our model's prediction of the response $\hat{y}$.

First, we standardize the output $c' = 2(c - 0.5)$ which then enters the argument of a logistic sigmoid function through the polynomial $Z = \omega_0 + \omega_1 c' + \omega_2 c'^3$.

$$\hat{y} = \frac{1}{1 + \exp(-Z)} \quad (5)$$

As we assume symmetry, only odd powers of $c'$ are used. In other words, the distorted confidence estimate $\hat{y}$ should lead to the same decision confidence regardless of whether the estimated majority is blue or red.

This function is flexible and able to approximate a wide range of distorted reports including the identity mapping and various forms of probability distortion[53,54]. It only accounts jointly for all effects which affect the final judgment. Other systematic deviations during confidence estimation which are conditional on a subset of the input space can only be partially accounted for, e.g., deviations for extreme values of the sample proportion.

**Ideal observer for Experiment 2.** The ideal observer model (see Fig. 3c) we describe here makes use of the generative process described in the main text and Fig. 3a, b. It updates a probability distribution over the observations of all in-block trials and their respective latent variables $(\mu_1, \ldots, \mu_T)$ up to the current trial $T$. The parameters $(v_1, v_2)$ defining the block tendency are part of the generative structure and assumed to be known. Consequently, inference amounts to an updating of the distribution over the latent variables through a calculation of the posterior distribution conditional on the observations (We identify the distributions by their respective arguments and e.g., write $p(D|\mu)$ for the distribution over the sufficient statistics of the sample. We often use the abbreviation $D = (q, N)$ for the observations, omitting parameters and index according to in-block trials $t$) as

$$p(\mu_1, \ldots, \mu_T, b | D_1, \ldots, D_T) \propto p(b) \prod_{t=1}^{T} p(D_t | \mu_t) p(\mu_t | b) \quad (6)$$

The current trial is labeled $T$, and $p(b)$ is the prior probability for block type $b$ ($p(b = 0) = p(b = 1) = 0.5$). Note that the probability distributions related to one block are independent to observations from previous blocks: in contrast to change-detection task paradigms, the model has explicit knowledge of when a new context start, and does not have to infer it. We checked in a control analysis that responses were only influenced by response in the same block but were not contaminated by responses in the previous block (Supplementary Fig. 9). This showed that subjects indeed incorporated the block structure into their inference process. The same knowledge was incorporated into heuristics models (see below) as well.

We would like to compute the probability of a blue latent trial majority, namely that $\mu_T$ is larger than 0.5. For this purpose, all variables relating to previous trials which are not of interest must be integrated out.

$$p(\mu_T \geq 0.5 | D_1, \ldots, D_T) = \frac{1}{\psi} \sum_{b=\{0,1\}} \int_{0.5}^{1} p(D_T | \mu_T) p(\mu_T | b) d\mu_T$$
$$\cdot p(b) \prod_{t=1}^{T-1} \int_{0}^{1} p(D_t | \mu_t) p(\mu_t | b) d\mu_t \quad (7)$$

The constant $\psi$ ensures normalization and can be recovered analytically as shown below. Because of conditional independence given the block tendency $b$, the high-dimensional distribution factorizes so that only one-dimensional integrals over the latent variables of previous trials must be performed. Examining the graph structure (see Fig. 3), we see that they may be considered messages $m_t(b)$ which are passed upwards to update the block-level variable $b$.

$$m_t(b) = \frac{1}{\psi_{m_t}} \int_{0}^{1} p(D_t | \mu_t) p(\mu_t | b) d\mu_t. \quad (8)$$

For proper normalization $\psi_{mt}$, they are themselves probability distributions that convey bottom-up evidence for the block tendency variable $b = \{0, 1\}$ based on the observations $D_t = (q_t, N_t)$. These bottom-up messages from previous trials within a block are integrated to update the belief $M_T(b)$ about the block tendency $b$ prior to

trial $t$ through point-wise multiplication and proper renormalization $\psi_M$.

$$M_T(b) = \frac{1}{\psi_M} p(b) \prod_{t=1}^{T-1} m_t(b) \quad (9)$$

As more evidence is gathered (trials), more factors can be absorbed into the belief about $b$ without having to store data from all previous trials independently as it is efficiently encoded in $M_T(b)$. Subsequently, this knowledge serves as top-down constraint on future inferences on the trial level. Consequently, to derive the probability of a blue trial majority on the current trial, the integration of momentary evidence (Eq. (6)) can be expressed as

$$p(\mu_T \geq 0.5 | D_1, \ldots, D_T) = \frac{1}{\psi} \sum_{b=\{0,1\}} M_T(b) \int_{0.5}^{1} p(D_T | \mu_T) p(\mu_T | b) d\mu_T \quad (10)$$

Proper normalization for the constants $\psi$, $\psi_M$ and $\psi_{mt}$ can be obtained analytically (see Supplementary Methods).

**Heuristic models to estimate the block tendency.** Here we describe three heuristic models that humans could use to estimate the block tendency.

1.  Averaging model
    The computation of the optimal estimate of a blue block tendency from previous trials, $M_T$ in Eq. (9), requires marginalization over hidden variables and normalization, which could be computationally difficult. Instead, participants could resort to approximations or heuristics. For the first model, the heuristic averaging model, we assume that the estimate of a blue block tendency ($b = 1$) is approximated by computing the average of the presented fractions of blue samples $q_t = N_{Bt}/(N_{Bt} + N_{Rt})$ in the trials $t$ prior to the current trial $T$ ($T \geq 2$).

    $$M_T^{avg}(b = 1) = \frac{1}{T-1} \sum_{t=1}^{T-1} q_t \quad (11)$$

    This estimate neglects sample size and corresponds to the implicit assumption that the inferred airplane's passenger proportion of each trial is well captured by a point estimate, i.e., by its respective sample proportion[17]. The model gives the same weight to each trial and thus ignores the fact that some trials provide more information than others due to different sample sizes. As for the other models below, indifference is assumed on the first trial $M_{T=1}^{avg}(b = 1) = 0.5$. The way the heuristics top-down message $M_T^{avg}$ is integrated into the confidence estimation process is described below (see Flexible mapping capturing current and prior information integration).
2.  Tally model
    Similarly, this model computes a tally of all blue samples observed prior to the current trial $T$ versus the number of all samples observed in a block so far.

    $$M_T^{tly}(b = 1) = \frac{\sum_{t=1}^{T-1} N_{Bt}}{\sum_{t=1}^{T-1} (N_{Bt} + N_{Rt})} \quad (11)$$

    This corresponds to pooling the samples of all trials, as if they were drawn from a common population of unknown population proportion.
3.  Difference model

    The heuristic difference model considers the difference between the number of blue and red samples $d_t = N_{Bt} - N_{Rt}$ in every observed trial $t$ within a block as informative to establish a belief about the block tendency. Across trials, it is accumulated by computing ($T \geq 2$):

    $$M_T^{d}(b = 1; \omega) = \frac{1}{1 + \exp(-\omega \cdot \sum_{t=1}^{T-1} d_t/(T-1))} \quad (13)$$

    The logistic sigmoidal function ensures that the result always takes a value between zero and one and that it can be interpreted as a proper belief, as in the previous two approximations. The parameter $\omega$ adjusts the sensitivity to the sample-difference statistics $d_t$ and can be determined by a fit to behavioral data.

**Flexible mapping capturing hierarchical integration.** This is a more flexible extension of the response mapping described before that can be used for the hierarchical learning task (Experiment 2). More concretely, we want to integrate any given prior belief $M$, not necessarily derived from a probabilistic model, with the momentary sample $D = (q, N)$ and map it onto the modeled response $(q, N, M) \mapsto \hat{y}$. As a mere function approximator, it is agnostic to the mechanisms that participants may use to combine information. Correspondingly, its parameters $\omega$ must be determined by a fit to the experimental data. Here, this process is approximated by a polynomial function $Z$ of the input $(q, N, M)$ that is fed into a logistic sigmoid as in Eq. (5).

$$Z = \omega_1 + \omega_2 q' + \omega_3 q' N + \omega_4 M + \omega_5 q'^3 + \omega_6 q'^3 N + \omega_7 NM' + \omega_8 M'^3 + \omega_9 NM'^3$$
$$(14)$$

The argument $Z$ contains only odd powers of $q$ and $M$ because we assume symmetry and no preference for estimating either red/blue majorities. Correspondingly, both quantities are standardized beforehand by the mapping $x' = 2(x - 0.5)$. As they are also independent from one another, no corresponding product terms are included.

Preliminary testing revealed that the inclusion of nonlinear terms is important to capture finer-grained patterns of behavior. The sample size $N$ is introduced into some terms to model its magnifying effect for the signed quantities $(q, M)$. We performed a weight normalization by the SD of each polynomial (for the input data) which was absorbed into the indicated weights $\boldsymbol{\omega}$. The particular choice of the terms in Eq. (14) balances flexibility with model complexity (and optimization for scarce behavioral data). We manually tested different parameterizations but did not find crucial differences for other reasonable choices of the mapping.

**Response distribution**. We assume that the probability of obtaining the behavioral confidence report $y_t$ on trial $t$ conditional on the data $\boldsymbol{d}_t$ and the model parameters is a Gaussian distribution truncated to the interval from zero to one $N_{[0,1]}(y_t|\hat{y}_t, \theta)$. The mean parameter of the normal distribution is set to the model prediction $\hat{y}_t$. The latter is denoted by $\hat{y}$ to distinguish it from the response $y$ of the participant which is formally represented by a draw from the response distribution to account for task-intrinsic behavioral variability beyond the variations captured by the model. The standard deviation (SD) parameter $\theta$ of the Gaussian is assumed to be constant and robustly estimated from the data (see Supplementary Methods). When analyzing the patterns of behavior produced by a fitted model (either a heuristic model or the optimal model with distortions), we computed the expected value of the mode under the truncated gaussian noise. Because of such truncated noise, the expected value is more centered than the noiseless model prediction:

$$\langle y_t|\hat{y}_t, \theta \rangle = \hat{y}_t + \theta \frac{N(\alpha) - N(\beta)}{\phi(\beta) - \phi(\alpha)}$$

where $\alpha = -\hat{y}_t/\theta$ and $\beta = (1 - \hat{y}_t)/\theta$

As our data might be contaminated by other processes such as lapses, we take precaution against far outlying responses. The response likelihood is calculated for all responses as

$$p(\mathbf{y}|\mathbf{d}_1, \dots, \mathbf{d}_T) = \prod_{t=1}^{T} (1 - \epsilon) N_{[0,1]}(y_t|\hat{y}, \theta) + \epsilon. \quad (15)$$

Additionally, to prevent isolated points from being assigned virtually zero probability we generally add a small probability of $\epsilon = 1.34 \times 10^{-4}$ to all. This corresponds to the probability of a point at four standard deviations from the standard normal distribution. For non-outlying points this alteration is considered negligible. To avoid singularity problems common to fitting mixture models, we constrained the SD parameter $\theta$ to be larger than 0.01 during fitting.

**Inferential patterns for fitted block tendency**. The probabilistic model assumes that the block tendency from which the trial-by-trial (airplane) proportions $\mu$ are drawn is given by one of two skewed Beta-distributions (see Methods). By convention a 'blue' ($b = 1$) context is characterized by the block tendency Beta$(\mu|\nu_1 = 14, \nu_2 = 9)$ while the 'red' context ($b = 0$) is correspondingly denoted by Beta$(\mu|\nu_2, \nu_1)$. The two distributions are symmetric with respect to the block aligned trial majorities, $\tilde{\mu}_b = b \cdot \mu + (1 - b) \cdot (1 - \mu)$, which immediately follows from the property of the Beta distribution: Beta$(\tilde{\mu}_{b=1}|\nu_1, \nu_2) = $ Beta$(\tilde{\mu}_{b=0}|\nu_2, \nu_1)$. A variation of the optimal inference routine (Eqs. (7–9)) is used that allows for different values of the parameters $\nu_1, \nu_2$ governing the block tendency with the restriction that $\nu_1 \geq \nu_2$. In addition, the sigmoidal response mapping (Eq. (13)) is used to allow for nonlinear distortions of the output.

**Estimating model evidence**. The evidence that each participant's data lends to each model is derived from predictive performance in terms of the cross-validation log likelihood (CVLL). For training, we maximized the logarithm of the response likelihood (Eq. (15)). To maximize the chances of finding the global maximum even for non-convex problems or shallow gradients, every training run first uses a genetic algorithm and then refines its estimate with gradient based search (MATLAB ga, fmincon). The CVLL for each participant and model is summarized by the median of the logarithm of the response likelihood (Eq. (15)) on the test set across all cross validation (CV) folds (SI).

**Group level comparison**. Instead of making the assumption that all participants can be described by the same model, we use a hierarchical Bayesian model selection method (BMS)[55] that assigns probabilities to the models themselves. This way, we assume that different participants may be described by different models. That is a more suitable approach for group heterogeneity and outliers which are certainly present in the data. The algorithm operates on the CVLL for each participant $(p = \{1, \dots, P\})$ and each model $(m = \{1, \dots, M\})$ under consideration and estimates a Dirichlet distribution Dir$(\boldsymbol{r}|\alpha_1, \dots, \alpha_M)$ that acts as a prior for the multinomial model switches $u_{pm}$. The latter are represented individually for each subject by a draw from a multinomial distribution $u_{pm} \sim \text{Mult}(1, \boldsymbol{r})$ whose parameters are $r_m = \alpha_m/(\alpha_1 + \dots + \alpha_M)$. We use the CVLL and assume an uninformative Dirichlet prior $\boldsymbol{a}_0 = 1$ on the model probabilities. Later, for model comparison, exceedance probabilities, $p_{exc} = \int_{0.5}^{1} \text{Beta}(\alpha_i, \sum_{j \neq i} \alpha_j)$, are calculated corresponding to the belief that a given model is more likely to have generated the data than any other model under consideration. High exceedance probabilities

indicate large differences on the group level. We consider values of $p_{exc} \geq 0.95$ significant (marked with *) and values of $p_{exc} \geq 0.99$ very significant (marked with **).

**Regression for sample size dependence**. Separate regression analyses conditional on sample size $N$ are used to determine the slope of the psychometric curves of the confidence judgments in a blue trial majority over the sample proportion of blue samples $q$ (Figs. 1, 2, 6). For a given sample size $N$, we use a logistic sigmoid with a linear weight $\omega_N$ to relate the standardized sample proportion $q'_N = 2(q_N - 0.5)$ to the modeled response $\hat{y}$.

$$\hat{y} = \frac{1}{1 + \exp[-\omega_N \cdot q'_N]} \quad (16)$$

We note that with this parameterization unbiased judgments are assumed. Conditioning reduces the number of data points available for fitting. To avoid numerical singularities (sigmoid collapses to step function) due to finite data, we use the likelihood function (Eq. (15)) but with the truncated Gaussian replaced by a Gaussian. This choice effectively leads to weighted regression assigning less probability density to responses close to the extremes (e.g., a response of 1 is assigned ½ of the density due to spill-over of the Gaussian into $[1, \infty)$). In this (heuristic) scheme, outlying responses are given less importance which translates into higher stability of the weight estimate.

**Regression for previous trial weights**. To estimate the weight on the sample proportion of previously presented in-block trials on the current confidence estimate we perform a regression analysis (see Figs. 4e and 7a). Probabilistic integration of evidence for the block tendency $M$ (Eq. (9)) results in a nonlinear increase of aligned confidence with the number of previously observed trials which saturates due to normalization. Hence, as the relative contribution of each trial decreases as more trials are observed, we perform the regression analysis separately for different numbers $(2, \dots, T - 1)$ of predictors (previous trials).

$$\hat{y} = \frac{1}{1 + \exp[-\sum_{t=1}^{T-1} \omega_t \cdot q'_t]} \quad (17)$$

As before, we use a logistic sigmoid with a linear combination of standardized sample proportion $q'_t = 2(q_t - 0.5)$ of each previous trial $t$ to the modeled response $\hat{y}$. Again, this conditioning reduces the number of data points available for fitting ($570/5 = 114$ trials) from which up to four weights have to be determined. To avoid numerical singularities due to finite data, we use the likelihood function (Eq. (15)) but with the truncated Gaussian replaced by a Gaussian (see above).

**Evidence-opposing choices due to contradictory prior**. Evidence-opposing choices are a crucial prediction of the ideal observer model which occur when the prior belief overrides contradictory evidence from the current sample. If we e.g., record a response that reports a blue majority while the sample majority is red, we call this an evidence opposing choice (confidence judgment). This can be attributed to an influence of an opposing prior belief or task-intrinsic response noise (input-independent). To avoid biased estimates because of the latter, the analysis is conditional on trials that on average provide opposing evidence to the sample. We only used trials whose aligned sample proportion is smaller than 0.5 as it opposes the tracked prior belief (on average).

Crucially, in Experiment 1, we found that noise basically does not lead to evidence opposing choices (see Supplementary Methods). Nevertheless, we make a conservative estimate by comparing behavior to a model whose evidence opposing choices just result from noisy responses in the absence of any prior belief tracking. This reference model $\hat{y} = \tilde{q} + \epsilon$ just reports the aligned sample proportion $\tilde{q}$ plus independent noise $\epsilon$ drawn from a truncated Gaussian distribution of standard deviation SD $= 0.1$.

**Binning for visualization and analyses**. To impose minimal constraints on data for visualization (see Figs. 5–7), we plotted the responses by grouping them into approximately equally filled bins across participants. The number of bins was manually chosen to achieve an appropriate trade-off between resolution and noise of the estimated bins values. Importantly, this only affects visualization. Unless stated otherwise, the underlying ungrouped data is used for testing. The conditional curves in Figs. 6b, c were determined by the cumulative quantiles $Q$ of the sample size distribution (many $\geq Q(0.6)$, few $< Q(0.4)$) and (many $> Q(0.5)$, few $\leq Q(0.5)$) respectively.

**Reporting summary**. Further information on research design is available in the Nature Research Reporting Summary linked to this article.

## Data availability
The data that support the findings of this study are available as Supplementary Data.

## Code availability
The code for data analysis is available publicly at https://github.com/pschustek/empirical_priors.

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

## Acknowledgements

P.S. was supported by a FI-AGAUR scholarship of the Secretariat for Universities and Research of the Ministry of Business and Knowledge of the Government of Catalonia and the European Social Fund. R.M.-B. is supported by BFU2017-85936-P and FLAGERA-PCIN-2015-162-C02-02 from MINECO (Spain) and Howard Hughes Medical Institute (HHMI, ref 55008742). A.H. is supported by the Jovenes Investigadores Grant from Spanish Mineco (PSI2015-74644-JIN). This work was supported by CERCA Programme/Generalitat de Catalunya. The funders had no role in study design, data collection and analysis, decision to publish or preparation of the manuscript. We are highly thankful to Alessia Cavallo for supporting experimental data acquisition.

## Author contributions

P.S. and R.M conceived the study. P.S. designed the task, ran the experiments, devised and implemented the models. P.S. and A.H. performed the data analyses. P.S., A.H. and R.M. discussed the results and wrote the manuscript.

## Competing interests

The authors declare no competing interests.
