## [Peer Review File · Nature Communications]

Reviewers' comments:

Reviewer #1 (Remarks to the Author):

In their manuscript, "Human confidence judgments reflect reliability-based hierarchical integration of contextual information", Philipp Schustek and Rubén Moreno-Bote examine how subjects make inferences when both the data and the source of the data are unreliable. In this setting, the rational behavior is to solve the problem hierarchically – use the evidence from previous observations to infer the source, and use the inferred source to better evaluate future observation. By probing subjects' confidence in relation to an ideal observer model, the authors conclude that humans rely on a graded belief about the source, and use that to make probabilistic judgments about observations.

The topic is timely and has been the focus of a number of recent papers. The key innovations in this work are 1) the task instructions were simple and intuitive, and 2) the experimental design afforded direct manipulation of reliability. Broadly speaking, the experiments were designed and implemented well, the modeling is of high quality, the data is solid, and the paper has the potential to be a strong contribution. However, results are not convincing unless several major shortcomings are addressed.

1. In many related studies, human performance deviate from probabilistic inference reported in this paper. The authors acknowledge this difference, and argue that the reason is that they explained the computational demands of their task in intuitive sense instead of using abstract instructions. While this may be the case, no evidence in support of this is provided. This is a critical and bears on the generality of the conclusions. Is the observation about probabilistic inference specific to this task or does it generalize? If it generalizes, then the authors needs to show that the same task with abstract instructions would not lead to probabilistic inference (i.e., negative control), or show that a results from a previous experiment that did not find evidence for probabilistic inference would change if the instructions are provided intuitively (i.e., positive control). Currently, I have no way of knowing if these results would generalize.

2. Based on our reading, a fundamental shortcoming of the paper is the block-design in Experiment 2. The block size seems to be fixed to 5 trials. This means that subjects may be well aware of the change points. If so, it seems inconceivable that subjects did not take the hazard function into account. However, the modeling seems to have completely ignored this variable. That is, the ideal observer is formulated as if it has no knowledge of the block design. This has to be remedied, and the behavior has to be compared with models that do take the hazard into account. Incorporating the hazard, could change many aspects of the results including the expected weight of previous evidence, the history-dependent deviations from ideal observer and various other conclusions.

3. The methods is somewhat opaque about the way feedback was provided (this has to be remedied), but our best inference is that subjects receive trial-by-trial feedback on their choice behavior (but not their confidence judgment). Is this the case? If so, the analyses and modeling has to be completely revised as it seems like this aspect is not taken into account. Let me explain why this is a major problem (although I am sure authors already know). Imagine a trial with 6 reds and 3 blues. Let's say the subject says red and received negative feedback. If the subject were to ignore the feedback, the sample would be consistent with a red context, but if s/he does, then this would be pretty good evidence for a blue context. In other words, taking the binary feedback into account could have strong influence on the subject and the model. This seems to have been neglected (or the explanations are very unclear).

4. If feedback was provided, some of the key analyses have to be revised. For example, the analysis of confidence as a function of trial in block has to be redone such that the number of preceding feedback are comparable. More feedback (both positive and negative) can and does

alter confidence regardless of the nature of the stimulus.

5. The subject's behavior deviate systematically from the ideal observer. That's not a problem but the fact authors ignore basic features of the data is not good. For example, they claim that subjects systematically overweight the current evidence (i.e., sometimes their decision goes against what optimal integration prescribes) but the paper did not attempt to reconcile this discrepancy between the model and behavior. This result indicates that the weight on current evidence on the inference about context is significantly larger than previous ones. If so, this needs to be characterized. Importantly, this goes against one of the central claims in the paper that the evidence is weighted equally. Rather, it seems that all previous trials were weighted similarly and differently from the current trial. Please quantify and discuss this point and show the behavior of a modified observer that discounts previous samples with respect to the current one.

6. There is no mention of single subjects almost anywhere in the paper. It is therefore, unclear if the results hold for all subjects, for some subset of subjects, or various features were mixed across subjects. The paper needs to reject the alternative hypothesis that the results are due to averaging.

Minor

1. I think some of the overstatements ("ubiquitous probabilistic representation") and not helpful. The data shows reliability-based reasoning not probabilistic representation. Using "reliability" to describe your results is a more accurate characterization of the results.

2. Statement not clear: "For all models, ... passed through a logistic function ... heuristic estimators." Please add page and line numbers to facilitate referencing.

3. Clarify the protocol about feedback. Explain the time out procedure more clearly. Explain the logic behind training sessions. Why did you use training blocks, and how did you verify that their performance was stable after the completion of training.

4. It would have been more powerful to analyze the results with respect to $m(b)$ and $M(b)$ as opposed to marginalizing the effects across binary sample sizes and various other marginalization. It is not clear to me why the authors did not use variables from the model to more concretely test their hypotheses.

5. Please don't overstate the results. In Discussion, it says "the match between observed and probabilistic inference patterns..." There are numerous deviations from the probabilistic inference pattern. It is a nice piece of work and the fact that humans cannot do the task ideally is not a problem. It is perfectly fine to state the results as is.

6. Range effect on the confidence line used for report may influence the results. Please discuss why this is not a concern or address it.

7. Can you please add careful analysis of behavior with respect to the hazard. For example, please show behavior for the first trial after each transition. Is there an effect of previous trial? There shouldn't be since the block size is known. Also carefully analyze the confidence measurement based on feedback as well as stimulus. In this type of experiment, it is really important to know the extent to which subjects rely on cumulative feedback versus inferred stimulus probabilities.

8. The authors highlight the importance of their design in terms of independently manipulating reliability (i.e., sample size). I really liked this manipulation, but the paper does not clearly explain why is this important. It says that previous studies did not do that but does not clarify why not having done that makes the conclusion from previous studies (e.g., Purcell and Kiani) any less believable. What is wrong with changing stimulus strength or duration? Indeed, changing stimulus

duration for a continuous variable seems analogous to sample size in for a discrete variable. Is this a validation of the previous work or does the independent manipulation of reliability addresses a scientifically important shortcoming in previous work?

9. Please clearly explain the symbols in all mathematical equations.

Reviewer #2 (Remarks to the Author):

The paper asks if humans can infer the latent hierarchical structure of the variations in an observed random variable. The paper describes two experiments in which evidence reliability is manipulated by varying sample size. The results show that participant's confidence reports are consistent with the author's proposed probabilistic inference model. The model outperforms conceptually simpler heuristic-based models. Preferred model's predictions are also refuted by the data in a couple of key comparisons.

The text is difficult to follow. The structure of the experiments are not easy to envisage and follow (see specific comments below). The paper's question is useful and interesting. It challenges the rather older previous notion that people do not factor in sample size when assessing the reliability of their evidence. However, more recent works (not mentioned in the paper; see below) have demonstrated evidence against blindness to sample size. The real conclusion of the paper should be: "it's complicated" as the data does deviate in interesting ways from the preferred model too.

The manuscript does not have page numbers.

Main comments:

1) The authors argue that the reason why their experiment is able to demonstrate sensitivity to sample size where previous works failed is because they have found a clever real-world metaphor (airplanes and football fans) for the cognitive problem of probabilistic inference. This would be a fantastic idea IF there was a control experiment to show that removing the metaphor from cover story (and keeping everything else, including detailed instructions, perceptual stimuli, response selection, reward schedule, practice sessions, etc.) reinstates blindness to sample-size in this exact same experiment. Because the structure of the current experiment is quite complex, expecting the subjects to show sample size insensitivity by default cannot be taken for granted.

2) Related to the above, the key findings about sample size sensitivity come from the early studies of Description-based tradition of risk elicitation literature. More recent works such as De Martino et al (Journal of Neuroscience 2017) have shown that when people take into account previous customer ratings of products, they do in fact factor in reliability and sample size in their updating of the product's value. As a result, it may also be the case that the probabilistic inference demonstrated here may be a consequence of experience-based design, repeated practice, or simply, more familiarity of the subject pool from 2018 to concepts and ideas of reliability compared to the far less computer-savvy participants of experiments in 1970.

3) As far as I could understand the paper, I was not convinced that the role of noise has been adequately addressed. Perceptual noise may have been more or less eliminated by allowing subjects to see the stimuli as long as they want. But noise in mathematical cognition (enumeration) is not the same as perceptual noise. So one would expect that there should be more noise in larger numerosities and smaller differences. Comparing model predictions and data in (eg figure 2A), data deviate considerably from predictions for larger sample sizes in the flanks of the sigmoid indicating lapse. There seems to be a fixed error that persists even when the task becomes very simple and that is not addressed or explained in the paper as far as I could see.

4) Some key aspects of experimental procedures are not clear. For example, what counts as

“correct” answer in Fig S1A? This relates critically to the experiment’s incentive structure that involves maximising “score” (page 1 of Methods). And when we read that the subjects were given feedback about their confidence in practice trials, I am not sure I understand how this is done. A single trial is either correct or not. There is no graded outcome. So there is no sense in which confidence can be modified to become better. So, how could we give feedback about confidence on a trial-by-trial basis?

5) It is great that the preferred model is compared with conceptually simpler models that use heuristics. However, intuitively, it seems like all of the models make qualitatively similar predictions about the experiments and there is only a quantitative difference between how much they agree with the data. Ideally one would have preferred the paper to include an experiment testing the DIVERGING predictions of the probabilistic and heuristic models. There is very little presentation of the predictions of the heuristic models in the paper and at least supplementary material could include this information.

Reviewer #3 (Remarks to the Author):

The manuscript reports two experiments investigating the influence of evidence strength, sample size and base rates on a probabilistic judgment task with confidence ratings. The results suggest that participants are sensitive to all of these factors, as reflected in their choices and particularly their confidence in these choices. The authors interpret the results as evidence that people perform probabilistic inference that integrates hierarchical information sources.

The manuscript has several important strengths. The research addresses topics of considerable current interest. The experiment is carefully designed and rigorously analyzed. Alternative explanations are considered and quantitatively evaluated against the main hypotheses. The manuscript is clearly written. My main concerns and criticisms are detailed below, followed by more minor queries and comments.

1) I am not persuaded that the results lead to new or surprising conclusions. Three ingredients seem sufficient to explain the results, which are (1) sensitivity to evidence strength, (2) sensitivity to sample size, and (3) sensitivity to base rates. It doesn’t strike me as a priori unexpected that people would be sensitive to all three factors. The task itself reinforces these effects through careful instructions and trial feedback (as discussed further below). Given this, what is proven by showing sensitivity to these factors? Put another way, if participants are given a probabilistic task together with reasons (instructions, incentives, feedback) to perform it well, isn’t their behavior almost guaranteed to be explained better by a probabilistic model than implausible non-probabilistic alternatives? Is there any reason to suppose people couldn’t learn the task?

2) Related to the first point, I’m not convinced by the conclusion that “our participants typically follow a probabilistic inference approach” (p.7). The authors make a strong version of this claim, which is that participants are representing probability distributions and transforming them lawfully according to probability theory, rather than a weaker claim that people are approximating probabilistic inference via other (e.g. heuristic) calculations. However, their model comparison doesn’t address this, only ruling out non-probabilistic models (e.g. a ratio model that is insensitive to sample size, and a simple counting model that is not probabilistic). Support for the “true inference” conclusion comes from in principle arguments (e.g. the difficulty of estimating uncertainty about latent variables, p.18). They didn’t need to collect data to make these arguments. And doesn’t the data suggest systematic deviation from predictions of the optimal model (e.g. systematic distortion away from extreme probabilities) that have often been taken to suggest the operation of heuristics rather than true probabilistic inference (e.g. in behavioral economics)? Many systematic deviations from the optimal model are not commented on (e.g. in

Figure 2a, why are participants systematically underconfident with extreme samples? In Figure 2b, why is the confidence slope so much higher than expected with sample size 13?). It also weakens their arguments that the analyses exclude participants who do not conform to a probabilistic model (a participant in Experiment 1 who seemed to use a counting strategy, and a participant in Experiment 2 who ignored base rates). I'm not sure it's reasonable to treat non-probabilistic behavior as "non-compliance" while interpreting probabilistic-like behavior as theoretically significant.

3) Because participants are given feedback on every trial, they could in principle use this experience to learn the optimal model, or simpler regularities like prediction accuracy being better with bigger sample sizes. Therefore, it would be interesting to see analyses of possible learning effects in the experiment, such as further from optimal confidence early in the experiment. A stronger test would be to collect data in another experiment without feedback on every trial, as a strong test that people are "naturally" capable of the kinds of probabilistic inference required in the task. For learning at a more local level, it would be worth checking if surprising feedback (i.e. that contradicts momentary evidence) leads to reduced confidence on the next trial, even in Experiment 1 where the trials are independent.

Minor comments

p.4. It would be helpful to clarify in the main text what feedback is provided to participants.

p.6. I wondered whether participants in Experiment 1 ever responded against the momentary evidence presented.

p.7 "passed thought" -> "passed through"

p.7. Please give details of the heuristic models in the Methods, in particular the equation(s) governing how the value NB - NR is converted into a probability value (confidence). It would also be useful in the Supplementary Materials to show plots (like those in Figure 1c and Figure 4a) showing the predictions of these alternative models, to give the reader a clear sense of where their predictions are violated.

p.7. It would be helpful to state explicitly in the main text that there were only two contexts and that these were the same throughout the experiment (not, for example, a range of possible contexts), and whether participants were told this as well.

p.12. "The gradual increase shows how nuanced the representation of the contextual variable is as there is no thresholding nor any sign of categorical representation." This is true across participants, but is it true at the level of individual participants' curves?

p.14. "Apparently, there are no signatures of temporally selective evidence integration". This statement seems to affirm the null hypothesis. And it does so in the face of a consistent trend towards the most recent trial (t-1) having a larger weight than the previous one (t-2). I didn't fully understand the description of the regression analyses in the Methods, but it doesn't seem like they were attuned to pick up this regularity (separate regressions for different trial numbers and/or coding according to trial in block rather than trial relative to the current one).

p.15. It would be useful to add to the Supplementary Materials a plot of the proportion of evidence-opposing choices in Experiment 2, e.g. as a function of sample size and/or proportion of samples.

Responses to reviewers' comments

We would like to thank the three reviewers for their very valuable comments that have pointed out multiple directions of how to consolidate our manuscript. The major changes we have performed are:

- Improving the clarity of the manuscript overall. In particular we have explained in more detail how feedback was provided to our participants and the reasons why we doubt they could play a role in shaping their response
- Showing that our participants' responses are better accounted for by the optimal model than by heuristics, not only quantitatively but also qualitatively: in both experiments there are some features of participant behavior that are only reproduced by the optimal model.
- Investigating in more depth that while the optimal model and participant behavior differ on several quantitative points, the two can be perfectly reconciled when taking into account distortions in participant response due to either a slightly different prior (Figure 8) or a distorted mapping from the estimate onto the final motor response (Figure S1).

We believe that our interpretations are now more clearly outlined and more strongly backed by the analyses (there are 7 additional supplementary figures). We hope this revised version will address the reviewers' concerns.

Reviewer #1 (Remarks to the Author):

In their manuscript, "Human confidence judgments reflect reliability-based hierarchical integration of contextual information", Philipp Schustek and Rubén Moreno-Bote examine how subjects make inferences when both the data and the source of the data are unreliable. In this setting, the rational behavior is to solve the problem hierarchically – use the evidence from previous observations to infer the source, and use the inferred source to better evaluate future observation. By probing subjects' confidence in relation to an ideal observer model, the authors conclude that humans rely on a graded belief about the source, and use that to make probabilistic judgments about observations.

The topic is timely and has been the focus of a number of recent papers. The key innovations in this work are 1) the task instructions were simple and intuitive, and 2) the experimental design afforded direct manipulation of reliability. Broadly speaking, the experiments were designed and implemented well, the modeling is of high quality, the data is solid, and the paper has the potential to be a strong contribution. However, results are not convincing unless several major shortcoming are addressed.

1. In many related studies, human performance deviate from probabilistic inference

reported in this paper. The authors acknowledge this difference, and argue that the reason is that they explained the computational demands of their task in intuitive sense instead of using abstract instructions. While this may be the case, no evidence is provided in support of this. This is a critical point and bears on the generality of the conclusions. Is the observation about probabilistic inference specific to this task or does it generalize? If it generalizes, then the authors need to show that the same task with abstract instructions would not lead to probabilistic inference (i.e., negative control), or show that a result from a previous experiment that did not find evidence for probabilistic inference would change if the instructions are provided intuitively (i.e., positive control). Currently, I have no way of knowing if these results would generalize.

We thank the three reviewers for noting the unnecessary boldness of our conclusions. At this point, the relationship between the type of instructions (abstract vs intuitive) and the nature of inference (probabilistic vs non-probabilistic) is a mere speculation that should be tested with control procedures as suggested by the reviewer. Such a point is not central to our study: our message is that under “some conditions” human subjects will spontaneously use hierarchical probabilistic inference. Delineating the precise conditions under which this occurs is beyond the scope of this study. Nevertheless, we thought that the speculative link with the instructions is an hypothesis worth discussing as it could reconcile our results with previous studies showing opposite effects.

We have changed the verb from “*believe*” to “*speculate*” in the sentence “*We speculate that the success of our participants in ‘understanding’ the hierarchical structure of the task is the result of the way the task has been framed and communicated.*” (L583)

We have also added the following sentence in the same paragraph: “*The existence of such framing effects onto the algorithmic nature of perceptual inference mechanisms should be tested in a future experiment.*” (L592-593)

2. Based on our reading, a fundamental shortcoming of the paper is the block-design in Experiment 2. The block size seems to be fixed to 5 trials. This means that subjects may be well aware of the change points. If so, it seems inconceivable that subjects did not take the hazard function into account. However, the modeling seems to have completely ignored this variable. That is, the ideal observer is formulated as if it has no knowledge of the block design. This has to be remedied, and the behavior has to be compared with models that do take the hazard into account. Incorporating the hazard, could change many aspects of the results including the expected weight of previous evidence, the history-dependent deviations from ideal observer and various other conclusions.

There was a misunderstanding, induced by our writing, about the nature of the inference process in the optimal and heuristics models, and thus we apologize for it. As described in the Methods section devoted to the hierarchical model, indices are meant “*according to in-block trials t* ”. Thus the inference process runs independently within each block, which is equivalent to saying that models do incorporate the knowledge of when a new context begins. Our task design clearly conveyed to the participants the start of a new block and the progression through its trials to the participant. This task contrasts with change-point detection (e.g. Glaze et al. eLife 2015) where only (at most) the hazard rate is known and the timing of changes of context (i.e. beginning of a

new block) must be inferred. This is now further specified in the Methods section (L580-6) as follows:

“Note that the probability distributions related to one block are independent to observations from previous blocks: in contrast to change-detection task paradigms, the model has explicit knowledge of when a new context start, and does not have to infer it. We checked in a control analysis that responses were only influenced by response in the same block but were not contaminated by responses in the previous block (Figure S9). This showed that subjects indeed incorporated the block structure into their inference process. The same knowledge was incorporated into heuristics models (see below) as well”

3. The methods is somewhat opaque about the way feedback was provided (this has to be remedied), but our best inference is that subjects receive trial-by-trial feedback on their choice behavior (but not their confidence judgment). Is this the case? If so, the analyses and modeling has to be completely revised as it seems like this aspect is not taken into account. Let me explain why this is a major problem (although I am sure authors already know). Imagine a trial with 6 reds and 3 blues. Let's say the subject says red and received negative feedback. If the subject were to ignore the feedback, the sample would be consistent with a red context, but if s/he does, then this would be pretty good evidence for a blue context. In other words, taking the binary feedback into account could have strong influence on the subject and the model. This seems to have been neglected (or the explanations are very unclear).

We apologize about not being clear at all about this very important point, so we are thankful that you brought it up. In experiment 1, we provided a simple binary feedback to indicate whether the participant's decision (side of the response) matched the actual underlying passenger majority of the airplane (the larger generating probability for each trial). To address the reviewer's concern, we have run further analyses: *“In principle the correctness feedback could be used by subject to learn the mapping from stimuli to the probability of selecting the correct category. In practice however, subject behavior was found to be very stable from the first test trial and throughout the session (Figure S6).”*

Regarding experiment 2, it is important to remark (but we were not clear enough in the previous version) that we did NOT provide trial-to-trial feedback. We only provided feedback as an overall score at the end of each block for experiment 2 (at each pause, i.e. every 5 trials for experiment 1). This feedback was a pure score but was not directional (“too much red”) so it could not even be used to guide behavior in future blocks. It was only provided to keep subjects motivated throughout the session.

We have now updated the manuscript by making the feedback scheme more explicit at various points of the paper. We hope that this will be clearer.

Results for Experiment 1 (pp4-5, L119-120 and 131-139)

“As no feedback was given that instructed our participants how they ought to make their confidence reports, the experiments probe their internal capacity to estimate uncertainties. (...) Importantly, there was no direct feedback about normative confidence reports: participants received a binary feedback after each response, i.e. whether they correctly identified the latent passenger majority. In addition, indirect feedback was provided at regular pauses every five trials through some aggregated performance score based on the ideal observer which was solely intended to maintain our participants engaged in the task. While such feedback could in

principle be marginally used to adapt one's responses, participants did not seem to modify their responses accordingly: first, feedback was hardly indicative of the optimal policy (see Methods); second, participants performed the task well from trial one and did not improve over time (see Fig. S6a)."

Methods for experiment 1 (p27, L817-34):

"After each trial, the participant receives feedback about the correctness of his decision (whether the cursor was placed on the side corresponding to the underlying passenger majority) but no supervising feedback regarding his confidence estimate. In addition, a two second time-out was presented for incorrect decisions which is signaled by a horizontal 'progress bar' which linearly diminishes over time indicating the fraction of the waiting time left. During time-out, there is nothing a participant can do to proceed but wait. In principle, the correctness feedback could be used by participants to learn the mapping from stimuli to the probability of selecting the correct category. In practice however, subject behavior was found to be very stable from the first test trial and throughout the session (Figure S6).

Every five trials, a pause screen was shown which provided information about how many out of all trials had already been completed. To motivate engagement in the task, we gave motivational feedback as an average $\langle S \rangle$ of the score S (distance to optimal observer, see above) over the last 5 trials since the last pause. Such feedback was uninformative as to how subjects should change their behavior to improve their score: Because it averaged performance over 5 trials, it was very unlikely they could use to learn current mappings and shape future responses (see stability of participants behavior Figure S6). Additionally, they also received a time-out of a few seconds proportional to $1 - \langle S \rangle$. The overall rationale behind the time-out was to more strongly incentivize task engagement and prevent click-through."

Results for Experiment 2 (p8, L230-2):

"Importantly, in Experiment 2, there was no feedback about decision correctness of each trial's airplane majority, only an overall score after each block (see Methods and Figure S7b)."

Methods for Experiment 2 (p29, L906-13):

"In Experiment 2 and different from Experiment 1, no feedback nor time-out was provided after each trial. However, as in Experiment 1, every five trials, i.e. after each block in Experiment 2, participants were presented with a pause screen with a score based on the results of the last block and a time-out of a few seconds proportional to $1 - \langle S \rangle$. As described before, the purpose was mainly to engage participants with the task. That they may have used this extremely sparse and indirect information to somehow guide future responses in Experiment 2, seems even more unlikely than in Experiment 1 as participants already showed no signs of converging to the normative strategy over time there (figure S6b) where the task was less complex than in Experiment 2 with several hidden variables. "

4. If feedback was provided, some of the key analyses have to be revised. For example, the analysis of confidence as a function of trial in block has to be redone such that the number of preceding feedback are comparable. More feedback (both positive and negative) can and does alter confidence regardless of the nature of the stimulus.

As feedback was not provided on a trial-by-trial basis, we believe the reviewer comment does not apply. Again, we apologize for being so opaque about it in the first version of the manuscript.

5. The subject's behavior deviate systematically from the ideal observer. That's not a problem but the fact authors ignore basic features of the data is not good. For example, they claim that subjects systematically overweight the current evidence (i.e., sometimes their decision goes against what optimal integration prescribes) but the paper did not attempt to reconcile this discrepancy between the model and behavior. This result indicates that the weight on current evidence on the inference about context is significantly larger than previous ones. If so, this needs to be characterized. Importantly, this goes against one of the central claims in the paper that the evidence is weighted equally. Rather, it seems that all previous trials were weighted similarly and differently from the current trial. Please quantify and discuss this point and show the behavior of a modified observer that discounts previous samples with respect to the current one.

We suspect that, unfortunately, there may have been another misunderstanding here about the property of the optimal model. The model predicts optimal weighting of all previous stimuli but not equal weighting of previous and current stimuli. This is specified in the manuscript (p14, L410): "*A central prediction of the probabilistic model is that all previous trials should have equal influence on behavior on average across blocks*". We would expect equal weighting between previous and current stimuli only if subjects had to report their estimate of the current context. Here, they reported their confidence about the majority of passengers in the current airplane, and hence we expect stronger weighting of the current stimulus. This is what simulations show and also what was found in subject behavior.

Still, the reviewer is correct by noting a certain departure of subject behavior to the optimal model in that the weighting of previous stimuli, albeit being equal across trial lags, was lower than that expected by the optimal model (Figure 7a). We have developed a series of analytical approximations to the optimal model that provides a good estimate of the respective weighting of the previous and current stimuli (see SI section '*Analytical approximations*'). This analytical framework reveals that the weighting of previous stimuli depends on the shape of the prior for each context: the more asymmetric the prior is (the larger the difference between v_1 and v_2), the larger is the previous stimulus weighting. In addition, this is born out by fitting the parameters of the Beta prior distribution that we used to model the block tendency (SI). This can be understood as, if the context is known, then the more biased the associated distribution, the stronger bias should be applied onto the inference process. Thus the lower weighting of previous stimuli by our subjects would naturally emerge if they would use the optimal model with a slightly more symmetrical prior than the one used to generate the stimuli. Indeed, model fitting of the participant Beta prior distribution confirmed that their behavior could be explained by a more symmetrical prior than the optimal one. Moreover, such fitted model was able to almost perfectly match the behavior produced by participants (new Figure 8).

6. There is no mention of single subjects almost anywhere in the paper. It is therefore, unclear if the results hold for all subjects, for some subset of subjects, or various

features were mixed across subjects. The paper needs to reject the alternative hypothesis that the results are due to averaging.

All analyses reported in the study are performed at the single-subject level first before group-level statistics are reported. Still, the reviewer comment is valid in asking whether the reported effects can be observed already at the single-subject level, or there can be due to averaging. Most effects can actually be observed in individual subjects. We have added examples in Figures S2 (slope as a function of sample size in experiment 1), S8a (aligned confidence as a function of aligned inferred tendency in experiment 2), S4a and S7a (model comparison for experiments 1 and 2 respectively). In some other analyses though, the statistical power was too weak to detect reliable patterns in individual subjects.

Minor

1. I think some of the overstatements (“ubiquitous probabilistic representation”) and not helpful. The data shows reliability-based reasoning not probabilistic representation. Using “reliability” to describe your results is a more accurate characterization of the results.

We thank the reviewer for this comment, as indeed some claims about probabilistic interpretations were slightly overstated in the original manuscript. This has now been corrected. For example the last sentence of the abstract has been changed to: *“Our results reveal uncertainty-sensitive integration of information at different hierarchical levels and temporal scales of the environment”*. We hope the current phrasing will satisfy the reviewers.

2. Statement not clear: “For all models, ... passed through a logistic function ... heuristic estimators.”

We thank the reviewer for noting this. Indeed the sentence was not clear, and the typo (through->thought) did not help! We have now reformulated to :

“To account for possible distortions on the response and/or calibrations of heuristics estimators, all model estimates (either from the optimal or heuristics models) were passed through a logistic function that mapped the estimate onto the unity interval.” (L203-6)

Please add page and line numbers to facilitate referencing.

Page and line numbers have been added.

3. Clarify the protocol about feedback. Explain the time out procedure more clearly. Explain the logic behind training sessions. Why did you use training blocks, and how did you verify that their performance was stable after the completion of training.

We addressed all those issues by rewriting the sections about the feedback in the Methods and added clarifying paragraphs at several places in the main text: pp5-6 (Results Experiment 1), p8 (Results Experiment 2), p27 (Methods Experiment 1), p29 (Methods Experiment 2).

Training blocks were used to allow subjects to get familiarized with the task. This is now clarified with more details in the manuscript (p 29, L887-99): *“As for Experiment 1*

and prior to starting session 2, participants completed two very short training sessions. First, they were given 20 trials (4 blocks) with a strong and visually obvious block tendency (sample sizes {8, ..., 11}, block tendency Beta(15,7)). Then another 30 trials under slightly harder conditions (sample sizes {3, ..., 11}, block tendency Beta(15,7)). Importantly, these training sessions only permitted them to understand the structure of the reasoning task, such as the dependence between the variables, and get familiarized with the task environment in increasingly difficult conditions. We intentionally provided as little information as possible as to how they should respond. The important point was to make clear what the structure of the process was that generated the samples. Thus, we did not monitor their performance, nor give them any feedback about how specifically they should place the response cursor. They could, however, ask the experimenter to clarify the assumptions behind the task. We proceeded to the actual experimental session when our participants reported that they had ‘understood’ the task. The above mentioned procedure was clear enough to achieve that, and yet sparse enough not to reveal the normative response strategy against which we wanted to compare their behavior.”

4. It would have been more powerful to analyze the results with respect to $m(b)$ and $M(b)$ as opposed to marginalizing the effects across binary sample sizes and various other marginalization. It is not clear to me why the authors did not use variables from the model to more concretely test their hypotheses.

We have tried to employ a mix of model-based and model-free analyses to test how much subject behavior matched the hierarchical inference model. In this sense figure 5b computed the reported confidence as a function of $M(b)$, in aligned space. We have made this more explicit by adding the label on the x-axis “aligned inferred tendency $M(b)$ ”. Following the reviewer’s comment, we have also added a figure showing the confidence as a function of the message passed from the previous trial $m_{t-1}(b)$ (Figure S8b). Again, we see that subject behavior qualitatively follows the optimal behavior. This complements the model-free analysis of figure 6c where we plot the aligned confidence as a function of both the aligned proportion and the sample size of the previous trial. The figure is referred to in the results section describing figure 6c (p15, L449-50).

5. Please don’t overstate the results. In Discussion, it says “the match between observed and probabilistic inference patterns...” There are numerous deviations from the probabilistic inference pattern. It is a nice piece of work and the fact that humans cannot do the task ideally is not a problem. It is perfectly fine to state the results as is.

Again, we agree with the reviewer that some interpretations were overstated. We have toned down the interpretations throughout the manuscript (see also Minor point 1 above) which we believe are now more in line with what we have actually observed. The expression above has been changed to “*the similarity between observed and probabilistic inference patterns...*” (p 17, L523).

6. Range effect on the confidence line used for report may influence the results. Please discuss why this is not a concern or address it.

From a normative point of view, confidence defined as the probability of making a correct choice should be bounded (as any probability). Intuitively too it is quite common to give a confidence measure along a bounded scale, e.g. “I am 90% confident that...”. This is exactly this kind of report that we instructed participants to provide as a

response. Still of course participants were free to use any subjective scale to map their internal estimate onto the response, possibly introducing some distortion in the response. This was our motivation to introduce flexible response mapping in all models (optimal and heuristics) between the model estimate and the response (see Methods). This mapping was fitted to each individual behavior, and we found that these distortions allowed to account for a good portion of participant departure from the optimal model in both experiments (Figure S1). Beyond this, the mapping was linear to a first approximation in most participants, which is a further indication that their internal estimate was represented in a format akin to a normative confidence measure.

7. Can you please add careful analysis of behavior with respect to the hazard. For example, please show behavior for the first trial after each transition. Is there an effect of previous trial? There shouldn't be since the block size is known.

The reviewer is right that, despite the beginning of the blocks being explicitly communicated to the subjects, some subjects could have partially used such information, e.g. using previous block inferred context to bias their response. This would show as some residual influence of previous block stimuli on subject responses. We have now run a control analysis to check for this: we have used the same logistic regression approach of figure 7a, but including information from trials in previous block as extra regressors. The results unequivocally show that subjects responses were not influenced by stimuli from previous blocks, i.e. they incorporated the explicit information about hazard provided by block initiation. This has been included in the manuscript (p31, L982-986) and in new Figure S9.

“We checked in a control analysis that responses were only influenced by response in the same block but were not contaminated by responses in the previous block (Figure S9). This showed that subjects indeed incorporated the block structure into their inference process. The same knowledge was incorporated into heuristics models (see below) as well.”

Also carefully analyze the confidence measurement based on feedback as well as stimulus. In this type of experiment, it is really important to know the extent to which subjects rely on cumulative feedback versus inferred stimulus probabilities.

Again we stress the fact that subject could not use cumulative feedback to shape their confidence measurement. We hope this is clearer now in the manuscript.

8. The authors highlight the importance of their design in terms of independently manipulating reliability (i.e., sample size). I really liked this manipulation, but the paper does not clearly explain why is this important. It says that previous studies did not do that but does not clarify why not having done that makes the conclusion from previous studies (e.g., Purcell and Kiani) any less believable. What is wrong with changing stimulus strength or duration? Indeed, changing stimulus duration for a continuous variable seems analogous to sample size in for a discrete variable. Is this a validation of the previous work or does the independent manipulation of reliability addresses a scientifically important shortcoming in previous work?

Good comment. First, while stimulus duration can modulate the reliability of the signal, it is impossible to control externally for how long subjects will actually integrate information (they could use simply the first 200 ms).

Second and most importantly, in experiments where stimulus duration or strength is manipulated, uncertainty emerges because the sensory system records unreliable measures. The optimal observer should take this sensory noise into account, but the particular optimal inference model will depend on the nature of the sensory noise, which is still a field of research in neuroscience. By contrast, in our study, the unreliability stems mostly from the fact that the cause for what is projected on the screen (the majority in the airplane) is hidden. This source of noise is independent from the sensory system and can be quantified objectively. In other words our “optimal observer” is not defined with respect to some sensory noise, which makes it a more reliable reference to compare with participants behavior. We have included this discussion in the manuscript (p18, L549-55):

“For example when using stimulus duration and stimulus strength as an indirect proxy to control reliability ¹⁵, the way these manipulations affect reliability depends on the specifics of the sensory system and sensory noise. In our task, uncertainty emerged not from sensory noise but from a hidden cause for stochastically generated stimuli and the reliability of both levels could be directly and independently controlled through sample size, thus providing an objective measure of trial-to-trial reliability independent of the sensory system.”

Finally, the task operates with small sample sizes, where the expected effects are largest.

9. Please clearly explain the symbols in all mathematical equations.

We have added further details about all the symbols used throughout the manuscript.

Reviewer #2 (Remarks to the Author):

The paper asks if humans can infer the latent hierarchical structure of the variations in an observed random variable. The paper describes two experiments in which evidence reliability is manipulated by varying sample size. The results show that participant’s confidence reports are consistent with the author’s proposed probabilistic inference model. The model outperforms conceptually simpler heuristic-based models. Preferred model’s predictions are also refuted by the data in a couple of key comparisons.

The text is difficult to follow. The structure of the experiments are not easy to envisage and follow (see specific comments below). The paper’s question is useful and interesting. It challenges the rather older previous notion that people do not factor in sample size when assessing the reliability of their evidence. However, more recent works (not mentioned in the paper; see below) have demonstrated evidence against blindness to sample size. The real conclusion of the paper should be: “it’s complicated” as the data does deviate in interesting ways from the preferred model too.

We thank the reviewer for his sensitive remarks. Our responses to this general review are provided below with each of the detailed points.

The manuscript does not have page numbers.

Page number and line numbers have been added, we are sorry for not having done it earlier.

Main comments:

1) The authors argue that the reason why their experiment is able to demonstrate sensitivity to sample size where previous works failed is because they have found a clever real-world metaphor (airplanes and football fans) for the cognitive problem of probabilistic inference. This would be a fantastic idea IF there was a control experiment to show that removing the metaphor from cover story (and keeping everything else, including detailed instructions, perceptual stimuli, response selection, reward schedule, practice sessions, etc.) reinstates blindness to sample-size in this exact same experiment. Because the structure of the current experiment is quite complex, expecting the subjects to show sample size insensitivity by default cannot be taken for granted.

We thank the reviewer for this issue which was pointed out by all reviewers. We completely agree that this idea is so far just a speculation, that should be tested in control experiments. We believe that this is a non-essential point of our study, and have modified the manuscript to make this clearer. As stated in our response to reviewer 1 question 1, our message is that under “some conditions” humans subjects will spontaneously use hierarchical probabilistic inference. Delineating the precise conditions under which this occurs is beyond the scope of this study. Nevertheless, we thought that the speculative link with the instructions is an hypothesis worth discussing as it could reconcile our results with previous studies showing opposite effects.

We have changed the verb from “believe” to “speculate” in the sentence “*We speculate that the success of our participants in ‘understanding’ the hierarchical structure of the task is the result of the way the task has been framed and communicated.*” (p19, L583)

We have also added the following sentence in the same paragraph: “*The existence of such framing effects onto the algorithmic nature of perceptual inference mechanisms should be tested in a future experiment.*” (p19, L592-4)

2) Related to the above, the key findings about sample size sensitivity come from the early studies of Description-based tradition of risk elicitation literature. More recent works such as De Martino et al (Journal of Neuroscience 2017) have shown that when people take into account previous customer ratings of products, they do in fact factor in reliability and sample size in their updating of the product’s value. As a result, it may also be the case that the probabilistic inference demonstrated here may be a consequence of experience-based design, repeated practice, or simply, more familiarity of the subject pool from 2018 to concepts and ideas of reliability compared to the far less computer-savvy participants of experiments in 1970.

We thank the reviewer for pointing out this very relevant reference and suggesting alternative interpretations of our data. Indeed the hypothesis that probabilistic inference would be employed if the task bears some similarity with decisions processes routinely performed by the subject is very interesting. It is somehow related to our interpretation while stressing more the familiarity of the task rather than its intuitiveness. We now discuss this alternative hypothesis in the discussion (p19-20, L594-600):

“A related but slightly different hypothesis is that probabilistic inference would be shaped during lifetime experience by repeated exposure to choices between options that require integration between sources of varying reliability. In the lab, such probabilistic inference process would only be applied if the task bears some similarity with the problems already encountered by the subjects in their life. In support of such hypothesis, a recent study did find sample size sensitivity in how subjects updated product evaluation by learning about previous consumers’ ratings, which is a highly familiar task routinely performed in everyone’s everyday life.”

3) As far as I could understand the paper, I was not convinced that the role of noise has been adequately addressed. Perceptual noise may have been more or less eliminated by allowing subjects to see the stimuli as long as they want. But noise in mathematical cognition (enumeration) is not the same as perceptual noise. So one would expect that there should be more noise in larger numerosities and smaller differences. Comparing model predictions and data in (eg figure 2A), data deviate considerably from predictions for larger sample sizes in the flanks of the sigmoid indicating lapse. There seems to be a fixed error that persists even when the task becomes very simple and that is not addressed or explained in the paper as far as I could see.

This is a very sensitive remark. Based on subject binarized response, overall noise was estimated to be very small (in experiment 1 subjects click on the side with larger number of dots in around 97% of cases). But indeed noise can be decomposed into sources that are dependent on sample size (numerosity noise) and sources independent of sample size (that could be due to central noise or motor noise). We have now run an analysis that disentangle the two sources (see Supplementary Information and Figure S5a) and found that both were quite weak. Thus they probably had limited impact on the estimate reports of the subjects.

Second, our interpretations of the deviations in the flank of the sigmoid relate to distortions of the mapping of the confidence onto the decision scale, as well as the possible use of a different prior by subjects than the true one, rather than attentional lapses. Lapses would be present if subjects would sometimes respond to the side opposite to the dominant colour even if the difference in dot counts was very large, which has we have stressed was almost never the case. Rather subjects chose the correct side but tended to undershoot their response. By contrast subjects tended to overshoot when the optimal model produces a centered responses, i.e. they were overconfident for the most ambiguous stimuli. We specifically fitted parametric distortions of the optimal model output on individual responses (see Methods), and found that such distortions perfectly explained for both undershooting and overshooting in Experiment 1 (see updated Figure S1).

4) Some key aspects of experimental procedures are not clear. For example, what counts as “correct” answer in Fig S1A? This relates critically to the experiment’s incentive structure that involves maximising “score” (page 1 of Methods).

We apologize for not being clear about this point. Actually this plot was simply meant to compare participant behavior to the optimal policy by representing the average participant response as a function of the optimal model response (which was confusingly termed “proportion correct” here). We have now updated the figure and changed the terminology, we hope it will be clearer.

And when we read that the subjects were given feedback about their confidence in practice trials, I am not sure I understand how this is done. A single trial is either correct or not. There is no graded outcome. So there is no sense in which confidence can be modified to become better. So, how could we give feedback about confidence on a trial-by-trial basis?

Obviously the feedback scheme was not clearly explained, as all reviewers have raised doubts about it. We apologize for this. The way the score is computed has been slightly rephrased (p26, L763): *“The score $S = 1 - |y - y_{opt}|$ of a response y was computed based on the proximity to the optimal confidence report y_{opt} .”* We agree that a binary response can only be correct or not, but for graded confidence response such as in our paradigm we can measure the accuracy based on the distance to the optimal response. In other words we do not simply ask subjects to report the most likely dominant category, but to report as accurately as possible the probability that the stimulus would be sampled from one or the other distribution. This has been clarified in the manuscript (p26): *“As such, the overall score reflected the ability of the subject to correctly infer the probability that the observed stimuli would be sampled from one category or the other.”*

5) It is great that the preferred model is compared with conceptually simpler models that use heuristics. However, intuitively, it seems like all of the models make qualitatively similar predictions about the experiments and there is only a quantitative difference between how much they agree with the data. Ideally one would have preferred the paper to include an experiment testing the DIVERGING predictions of the probabilistic and heuristic models. There is very little presentation of the predictions of the heuristic models in the paper and at least supplementary material could include this information.

We thank the reviewer for this very valuable comment, which has prompted us to study in more detail different qualitative predictions of the different models.

The falsified predictions of the ‘difference’ and ‘ratio’ models of Experiment 1 and 2 are now plotted in Figure S3 and S10, respectively. In both cases, the best fit of each heuristic model to the experimental data is not capable of reproducing all qualitative patterns at once [if any at all].

To gain generality, we have derived mathematically a set of analytical approximations for each model. The approximations describe each of the characteristics of the model behavior (dependence of the slope of the psychometric curve on sample size of the current trial, on the proportion and sample size of previous trials, etc.) depending on the model parameters. From this we could extract reliable qualitative differences between the different models that were completely independent of parameter estimations:

- In experiment 1, the slope of the psychometric curve should not depend on sample size for the ‘ratio’ model, should depend linearly according to the ‘difference’ model, and sublinearly according to the optimal model, as was actually observed in subjects (figure S3). The analytical treatment also revealed that the difference model actually corresponds to the optimal model if the stimuli would be extracted from two fixed distributions (e.g. either 60% blue or 40% blue), corresponding to delta priors for the sample proportion. The fact that subjects’ behavior used a more complex strategy that

incorporates uncertainty at the sample proportion level (for each category) suggest that inference naturally takes into account uncertainty at various levels.

- In experiment 2, one different qualitative difference between the optimal model and the heuristics is that the belief about the current block context accumulated evidence across trials, while in all three heuristics models belief averages evidence from the different trials. This gives rise to different predictions as to how the influence of the overall proportion of blue samples in previous trials of the block changes depending on trial position. In the optimal model this influence increases with trial position as the belief accumulates over more trials, while the influence remains constant in other models. Figure S10 show that participants display the same pattern of response as the optimal model, clearly invalidating predictions from the heuristics models. Moreover, we also show that the 'averaging model' is by construction insensitive to the sample size of the previous trial (Figure S10f), unlike our participants (Figure S7c).

The analytical derivations have been added in Supplementary Information, and the results related to the new Supplementary Figures are described in the relevant sections of the main text. Altogether, we believe that these new analyses provide a stronger support in favor of probabilistic-like inference in our task, allowing to rule out on a firmer basis alternative heuristics models. Again we thank the reviewer for pointing towards this direction.

Reviewer #3 (Remarks to the Author):

The manuscript reports two experiments investigating the influence of evidence strength, sample size and base rates on a probabilistic judgment task with confidence ratings. The results suggest that participants are sensitive to all of these factors, as reflected in their choices and particularly their confidence in these choices. The authors interpret the results as evidence that people perform probabilistic inference that integrates hierarchical information sources.

The manuscript has several important strengths. The research addresses topics of considerable current interest. The experiment is carefully designed and rigorously analyzed. Alternative explanations are considered and quantitatively evaluated against the main hypotheses. The manuscript is clearly written. My main concerns and criticisms are detailed below, followed by more minor queries and comments.

1) I am not persuaded that the results lead to new or surprising conclusions. Three ingredients seem sufficient to explain the results, which are (1) sensitivity to evidence strength, (2) sensitivity to sample size, and (3) sensitivity to base rates. It doesn't strike me as a priori unexpected that people would be sensitive to all three factors.

While there is not one single result in the study that univokely rules out all other possible heuristics, the fact that all patterns of results were in line qualitatively and (most often) quantitatively with the optimal inference model provides a converging body of evidence in favour of probabilistic inference. It should be noted that while some of the effects had been reported in isolation previous studies, the study brings them into perspective by proposing a single framework that accounts for all of them: probabilistic hierarchical inference. Besides, sensitivity to sample size in abstract inference tasks is

still a matter of debate as there are several studies that have cast doubt on the general sensitivity to sample size and base rates [Kahneman, *Thinking, Fast and Slow*. 2011], or more generally the applicability of normative theories to describe cognition [Gigerenzer & Gaissmaier, 2011] at least if cognitive resource limitations are not accounted for [e.g. Gershman et al Science 2015]. Our study goes beyond evidencing that human participants are sensitive to the above-mentioned factors. We attempted to show that people are capable of using a principled approach to inference by showing that alternative accounts fail to reproduce behavior.

To clarify this point, we have added the following sentences in the introduction “*While some effects predicted by hierarchical probabilistic inference have been previously reported in isolation, no study has to our knowledge assessed thoroughly a body of behavioral predictions of hierarchical probabilistic inference and tested them against alternative heuristics model.*” and later on “*Overall, all the reported effects in both tasks were consistent quantitatively and qualitatively with the optimal inference model.*” (pp2-3, L67-70)

The task itself reinforces these effects through careful instructions and trial feedback (as discussed further below). Given this, what is proven by showing sensitivity to these factors? Put another way, if participants are given a probabilistic task together with reasons (instructions, incentives, feedback) to perform it well, isn't their behavior almost guaranteed to be explained better by a probabilistic model than implausible non-probabilistic alternatives? Is there any reason to suppose people couldn't learn the task?

Our manuscript was not clear about what feedback consisted in. This has created misunderstandings in all three reviewers, we apologize for this. As explained in more details in response to reviewer 1 question 2, we did not provide trial-to-trial feedback to the participants that would allow them to learn the mapping of the probabilistic model. In experiment 2, the feedback merely consisted in a cumulative score, designed to incentivize subjects to perform well throughout the session, but not informative enough to shape their response. In experiment 1, they also received binary feedback about the side of their choice, not the graded response. In fact, the task aimed at not reinforcing these effects by not allowing our participants to learn from their interaction with the task over the course of the experiment. Additionally, we have provided evidence that their performance did not improve with the number of completed trials (Fig. S6).

2) Related to the first point, I'm not convinced by the conclusion that “our participants typically follow a probabilistic inference approach” (p.7). The authors make a strong version of this claim, which is that participants are representing probability distributions and transforming them lawfully according to probability theory, rather than a weaker claim that people are approximating probabilistic inference via other (e.g. heuristic) calculations. However, their model comparison doesn't address this, only ruling out non-probabilistic models (e.g. a ratio model that is insensitive to sample size, and a simple counting model that is not probabilistic). Support for the “true inference” conclusion comes from in principle arguments (e.g. the difficulty of estimating uncertainty about latent variables, p.18). They didn't need to collect data to make these arguments.

We agree with the reviewer that the original manuscript somehow jumped too boldly to the conclusion that our participants use a probabilistic inference approach. A purely behavioral study cannot reach a definite answer about the nature of the representations used during the inference process. Mostly we can say that the behavior is compatible with probabilistic inference but could also be reached by some complex approximative heuristics. This was already acknowledged in the Discussion (p19, L563-78), but it is true that the tone in other parts of the manuscript was more assertive. We have modified the manuscript to better align our conclusions to our results. In particular, the sentence p6 was changed to *“the response patterns of our participants suggests a probabilistic inference approach”*.

We would also like to point out that while it is, of course, perfectly expected that the sample-size-insensitive ratio model fails on showing sample size effects, the heuristic models that estimate the block tendency in Experiment 2 (the averaging, the tally and the difference model) do in fact integrate information across trials. In addition to the model comparison procedure, we now also show that they do not match the behavioral patterns as closely as the more principled probabilistic inference approach (Fig. S10). Furthermore, we show that, under some assumptions, our heuristic models may be understood as an approximation to the optimal inference approach (see SI).

And doesn't the data suggest systematic deviation from predictions of the optimal model (e.g. systematic distortion away from extreme probabilities) that have often been taken to suggest the operation of heuristics rather than true probabilistic inference (e.g. in behavioral economics)? Many systematic deviations from the optimal model are not commented on (e.g. in Figure 2a, why are participants systematically underconfident with extreme samples?)

The reviewer rightfully notes some systematic deviations of participants' behavior from the optimal model. We believe that some, though not all, of these deviations can be understood within the framework of the optimal model. For example the fact that subjects tended to 'undershoot' their responses for extreme probabilities can be attributed to some distortions in subjects' mapping from the inferred probability to the motor response (see updated Figure S1). Such distortions seem to be independent of the central inference stage (reliability weighting by sample size), as they appear for all sample sizes.

In addition, distortions need not necessarily arise from the inference stage, but might e.g. stem from motor responses. For example, subject will take into account efficiency considerations to move the mouse in a repetitive task. Another deviation is the fact that participants of experiment 2 displayed a lower sensitivity to previous stimuli than the optimal model. This too can be reconciled within the framework of the optimal model, if subjects were using a less skewed beta prior than the actual one - i.e. would believe both blue (red) context are less biased towards blue (red) dots than what they actually are (see response to reviewer 2 question 5 for more details). In some, while these deviations existed, they could naturally be interpreted within the framework of probabilistic inference. However, within this framework, e.g. our formulation of the block tendency with a skewed Beta-distribution is a choice, but not expected to be general across participants. Hence, it is to some degree perfectly expected that our participants may make different assumptions about the distribution's structure. Naturally, inference

based on different distributions leads to systematically biased outputs with respect to our model.

Another factor might be distortion in estimation of the number of passengers of both colours, or noise in this estimation process (see also Fig. S5). The optimal model assumes noiseless perception of the numbers, but variability in the estimation of the number of passengers will inevitably introduce some departure of the behavior from the optimal model.

Finally, perceptual noise (numerosity noise or other forms of sensory noise) could play a detrimental role onto our participant behavior, although we estimated this contribution to be minimal (see revised analysis in SI section 'Sensory noise').

A paragraph is devoted in the Discussion (p 20) to these systematic deviations. We have developed it a bit further by including the impact of mapping distortions and sensory noise (L612-617): *"Beyond this differences in the central inference stage, there could be alternative sources of distortions in the conversion from the estimate into a motor report. Taking into account such distortions allowed to capture some other part of the departure of our participants' behavior to the optimal observer (Figure S1). By contrast, our participants behaviour was found to be little affected by numerosity or other forms of sensory noise (see SI)."*

In Figure 2b, why is the confidence slope so much higher than expected with sample size 13?).

Note the large variability across subjects in the estimated value for sample size 13, the deviation from the optimal model is actually not significant (two-sided t-test, $p = 0.21$).

It also weakens their arguments that the analyses exclude participants who do not conform to a probabilistic model (a participant in Experiment 1 who seemed to use a counting strategy, and a participant in Experiment 2 who ignored base rates).

Good comment. Our claim is not necessarily that ALL subjects use probabilistic-like inference, but rather that they can all potentially use it. In this experience we show that MOST subjects behavior is compatible with probabilistic-like inference, but as discussed in other parts many previous studies have already showed that in other circumstances subjects resort to different strategies. Here, our interpretation was that two subjects overall misunderstood the task instructions and thus their behavior was non-compliant with task's intention of tracking information from previous trials (see SI section 'Compliance with hierarchical task'). Importantly, no participant had been excluded on the basis of non-compliance to the probabilistic model (or any model we advocate) but according to a more fundamental standard whether they used information from previous trial at all. To address the reviewer's comment, we have run the same analyses without excluding them: now the only participant that was excluded is the one that did not complete the experiment (we also solved a numerical problem that gave aberrant model fits for one participant). All results were preserved qualitatively and quantitatively (all figures and statistical tests have been updated accordingly in the manuscript).

Besides, the Bayesian model selection readily provides the estimated probability that each subject would use the optimal model. The fact that this probability is large in both

experiments, even without subject exclusion, shows that most subjects behavior is compatible with probabilistic-like inference.

I'm not sure it's reasonable to treat non-probabilistic behavior as "non-compliance" while interpreting probabilistic-like behavior as theoretically significant.

As stated above, we believe that the responses are compliant with probabilistic-like behavior when the full model incorporates some very reasonable extra ingredients (distorted response mapping, skewed beta prior). Moreover, our original decision to leave out participants was based on the compliance to the hierarchical nature of the task, i.e. the tracking of information from previous trials (not necessarily in a graded manner). Not only the probabilistic inference model integrates information from previous trials, our heuristic hierarchical models do as well. Thus, participants whose behavior show (virtually) no dependence on previous trials are not as informative to arbitrate between our models; including or excluding them does not result in any bias towards on the model. Finally, as mentioned above, our claim is not necessarily that all participants always use probabilistic-like inference, but that it is in the repertoire of human cognitive abilities.

3) Because participants are given feedback on every trial, they could in principle use this experience to learn the optimal model, or simpler regularities like prediction accuracy being better with bigger sample sizes. Therefore, it would be interesting to see analyses of possible learning effects in the experiment, such as further from optimal confidence early in the experiment. A stronger test would be to collect data in another experiment without feedback on every trial, as a strong test that people are "naturally" capable of the kinds of probabilistic inference required in the task. For learning at a more local level, it would be worth checking if surprising feedback (i.e. that contradicts momentary evidence) leads to reduced confidence on the next trial, even in Experiment 1 where the trials are independent.

Such criticism was common to all reviewers. We therefore copy here the response made to reviewer 1.

We apologize about not being clear at all about this very important point, so we are thankful that you brought it up. In experiment 1, we provided a simple binary feedback to indicate whether the participant's decision (side of the response) matched the actual underlying passenger majority of the airplane (the larger generating probability for each trial). To address the reviewer's concern, we have run further analyses: *"In principle the correctness feedback could be used by subject to learn the mapping from stimuli to the probability of selecting the correct category. In practice however, subject behavior was found to be very stable from the first test trial and throughout the session (Figure S6)."*

Regarding experiment 2, it is important to remark (but we were not clear enough in the previous version) that we did NOT provide trial-to-trial feedback. We only provided feedback as an overall score at the end of each block for experiment 2 (at each pause, i.e. every 5 trials for experiment 1). This feedback was a pure score but was not directional ("too much red") so it could not even be used to guide behavior in future blocks. It was only provided to keep subjects motivated throughout the session.

We have now updated the manuscript by making the feedback scheme more explicit at various points of the paper. We hope that this will be clearer.

Results for Experiment 1 (pp4-5, L119-120 and 131-139)

“As no feedback was given that instructed our participants how they ought to make their confidence reports, the experiments probe their internal capacity to estimate uncertainties. (...) Importantly, there was no direct feedback about normative confidence reports: participants received a binary feedback after each response, i.e. whether they correctly identified the latent passenger majority. In addition, indirect feedback was provided at regular pauses every five trials through some aggregated performance score based on the ideal observer which was solely intended to maintain our participants engaged in the task. While such feedback could in principle be marginally used to adapt one’s responses, participants did not seem to modify their responses accordingly: first, feedback was hardly indicative of the optimal policy (see Methods); second, participants performed the task well from trial one and did not improve over time (see Fig. S6a).”

Methods for experiment 1 (p27, L817-34):

“After each trial, the participant receives feedback about the correctness of his decision (whether the cursor was placed on the side corresponding to the underlying passenger majority) but no supervising feedback regarding his confidence estimate. In addition, a two second time-out was presented for incorrect decisions which is signaled by a horizontal ‘progress bar’ which linearly diminishes over time indicating the fraction of the waiting time left. During time-out, there is nothing a participant can do to proceed but wait. In principle, the correctness feedback could be used by participants to learn the mapping from stimuli to the probability of selecting the correct category. In practice however, subject behavior was found to be very stable from the first test trial and throughout the session (Figure S6).

Every five trials, a pause screen was shown which provided information about how many out of all trials had already been completed. To motivate engagement in the task, we gave motivational feedback as an average $\langle S \rangle$ of the score S (distance to optimal observer, see above) over the last 5 trials since the last pause. Such feedback was uninformative as to how subjects should change their behavior to improve their score: Because it averaged performance over 5 trials, it was very unlikely they could use to learn current mappings and shape future responses (see stability of participants behavior Figure S6). Additionally, they also received a time-out of a few seconds proportional to $1 - \langle S \rangle$. The overall rationale behind the time-out was to more strongly incentivize task engagement and prevent click-through.”

Results for Experiment 2 (p8, L230-2):

“Importantly, in Experiment 2, there was no feedback about decision correctness of each trial’s airplane majority, only an overall score after each block (see Methods and Figure S7b).”

Methods for Experiment 2 (p29, L906-13):

“In Experiment 2 and different from Experiment 1, no feedback nor time-out was provided after each trial. However, as in Experiment 1, every five trials, i.e. after each block in Experiment 2, participants were presented with a pause screen with a score based on the results of the last block and a time-out of a few seconds proportional to $1 - \langle S \rangle$. As described before, the purpose was mainly to engage participants with the task. That they may have used this extremely sparse and indirect information to somehow guide future responses in Experiment 2, seems even more

unlikely than in Experiment 1 as participants already showed no signs of converging to the normative strategy over time there (figure S6b) where the task was less complex than in Experiment 2 with several hidden variables. ”

Minor comments

p.4. It would be helpful to clarify in the main text what feedback is provided to participants.

This was very much necessary indeed! We have now clarified the presentation of feedbacks in the manuscript, and hope it will avoid future confusions.

p.6. I wondered whether participants in Experiment 1 ever responded against the momentary evidence presented.

Yes, for a very small proportion of trials (3.05% on average across participants with a standard deviation of 2.72%). This was used to estimate the sensory noise (in the revised manuscript, we now dissociate the parts of the sensory noise that sample size dependent and sample size independent). See section ‘*Sensory noise*’ in supplementary material for further information.

p.7 “passed thought” -> “passed through”

Corrected, thank you.

p.7. Please give details of the heuristic models in the Methods, in particular the equation(s) governing how the value NB - NR is converted into a probability value (confidence).

We have added more details about the heuristics model, including the equation for the difference model.

It would also be useful in the Supplementary Materials to show plots (like those in Figure 1c and Figure 4a) showing the predictions of these alternative models, to give the reader a clear sense of where their predictions are violated.

The original manuscript showed that the optimal model provided a better quantitative account of subjects’ behavior (bayesian model selection), but did not quite address the question of qualitative accounts. We have remedied this by inserting a number of figures (Fig. S3 and S10) showing how predictions from the different heuristics model are invalidated by subject behavior. As explained in our response to reviewer 2, for both experiments, the best fit of each heuristic models to the experimental data is not capable of reproducing all qualitative patterns at once [if any at all].

To gain generality, we have derived mathematically a set of analytical approximations for each model. The approximations describe each of the characteristics of the model behavior (dependence of the slope of the psychometric curve on sample size of the current trial, on the proportion and sample size of previous trials, etc.) depending on the model parameters. From this we could extract reliable qualitative differences between the different models that were completely independent of parameter estimations:

- In experiment 1, the slope of the psychometric curve should not depend on sample size for the 'ratio' model, should depend linearly according to the 'difference' model, and sublinearly according to the optimal model, as was actually observed in subjects (figure S3). The analytical treatment also revealed that the difference model actually corresponds to the optimal model if the stimuli would be extracted from two fixed distribution (e.g. either 60% blue or 40% blue), corresponding to delta priors for the sample proportion. The fact that subjects behavior used a more complex strategy that incorporates uncertainty at the sample proportion level (for each category) suggest that inference naturally takes into account uncertainty at various levels.

- In experiment 2, one different qualitative difference between the optimal model and the heuristics is that the belief about the current block context accumulated evidence across trials, while in all three heuristics models belief averages evidence from the different trials. This gives rise to different predictions as to how the influence of the overall proportion of blue samples in previous trials of the block changes depending on trial position. In the optimal model this influence increases with trial position as the belief accumulates over more trials, while the influence remains constant in other models. Figure S10 show that participants display the same pattern of response as the optimal model, clearly invalidating predictions from the heuristics models. Moreover, we also show that the 'averaging model' is by construction insensitive to the sample size of the previous trial (Figure S10f), unlike our participants (Figure S7c).

The analytical derivations have been added in Supplementary Information, and the results related to the new Supplementary Figures are described in the relevant sections of the main text. Altogether, we believe that these new analyses provide a stronger support in favor of probabilistic-like inference in our task, allowing to rule out on a firmer basis alternative heuristics models. Again we thank the reviewer for pointing towards this direction.

p.7. It would be helpful to state explicitly in the main text that there were only two contexts and that these were the same throughout the experiment (not, for example, a range of possible contexts), and whether participants were told this as well.

We now specify in the Methods section (p28, L843-4): "*There were two possible contexts: one biased towards red passengers, the other towards blue passengers.*" Regarding instructions, as explained in SI, subjects were told "*that the tendency for 'red' or 'blue' airplane majorities changes unpredictably from city to city and does not favor either group.*" We did not provide further indications, e.g. if some cities could have more extreme biasing towards one colour than others.

p.12. "The gradual increase shows how nuanced the representation of the contextual variable is as there is no thresholding nor any sign of categorical representation." This is true across participants, but is it true at the level of individual participants' curves?

We thank the reviewer for the comment. There is also no thresholding at the individual level, as can be seen in the new figure S8. Behavior was very consistent across subjects. Subject responses correlated individually with aligned inferred tendency (Pearson correlation at $p < 0.05$) for 17 out of 24 participants.

p.14. “Apparently, there are no signatures of temporally selective evidence integration”. This statement seems to affirm the null hypothesis. And it does so in the face of a consistent trend towards the most recent trial (t-1) having a larger weight than the previous one (t-2).

I didn't fully understand the description of the regression analyses in the Methods, but it doesn't seem like they were attuned to pick up this regularity (separate regressions for different trial numbers and/or coding according to trial in block rather than trial relative to the current one).

We apologize for the unclear formulation of the analysis that we have realized. We are regressing the weight obtained from the regression analysis against trial position. We believe such analysis would capture if there was a tendency for larger or lower weights for the latest trials (respectively recency or primacy effect). This is now clarified (p14, L413-7): *“Accordingly, no significant trend could be evidenced through another linear regression analysis in which the previous trial index is used to predict the average weight of the previous trial on the aligned confidence (regression on the means across participants, separately for current trials position 3, 4, and 5: p-values 0.417-0.897 for trials with 2-4 previous trials respectively).”*

Following the reviewer's advice, we performed another analysis to further test for an apparent primacy effect. We performed a t-test between participant weights at trials t-2 and t-1, separately for the regression performed on trials at position 3 (i.e. comparing weights for position 1 and 2), for the regression performed on trials at position 4 (i.e. position 2 and 3), and at position 5 (i.e. position 3 and 4). None of the test were significant (p-values 0.08, 0.34, 0.38, uncorrected). When grouping weights from positions 3, 4, and 5, it did not either reach significance (p=0.060). So the visible trend does not seem to be significant. Note that a primacy effect would not be a major problem for our theoretical framework, as such “leak” in the integration process can stem from a belief by participants that the context has some non-null probability of switching within a block (Glaze et al, eLife 2015).

p.15. It would be useful to add to the Supplementary Materials a plot of the proportion of evidence-opposing choices in Experiment 2, e.g. as a function of sample size and/or proportion of samples.

Good suggestion, this has now been added as Figure S5b.

REVIEWERS' COMMENTS:

Reviewer #2 (Remarks to the Author):

I have now read the paper and, more importantly, the rebuttal. My impression is not positive.

All three reviewers found the paper's key interesting point to be the claim that in an ecologically intuitive context, people can do probabilistic, sample-size sensitive judgements. All three reviewers also independently and unanimously ask for a control experiment in which the same task with abstract instructions is presented to participants and previous insensitivity to sample size is replicated.

The authors' response is to wriggle out of this straightforward request. They argue that their point is that probabilistic inference is "possible" and at least some subjects under some conditions can do it. They also acknowledge that others (eg De Martino et al 2015) have indeed demonstrated sample size sensitivity and that others (eg Purcell & Kiani) have demonstrated hierarchical probabilistic reasoning in humans. All of this puts the reader in a difficult position: the paper is now reduced to a very restricted claim that has been previously demonstrated several times. It also retracts the very claim that all three reviewers found interesting about the paper (ie the role of intuitive context).

To me, the paper is now much less interesting. I do not find the claims bold or controversial anymore and pretty much agree with reviewer 3 that, anyone who runs a probabilistic learning task with clear enough instructions and adequate practice will demonstrate the same result WITHOUT adding anything new to our understanding of the cognitive basis of probabilistic inference in human mind and brain.

Reviewer #3 (Remarks to the Author):

Note: The editor asked me to evaluate the manuscript also in relation to Reviewer 1's comments.

The authors have provided a very responsive revision of the manuscript, including several new analyses and plots that expand and reassure on important points. The clarification about the nature of feedback (or lack of it) provided to participants addresses most of my comments and those of Reviewer 1, and the new model fit in Fig. 8 is convincing. Altogether the manuscript makes a convincing argument about an important and timely issue. It would nevertheless benefit from attention to a few remaining minor issues:

1) In my first review I highlighted the deviation from model predictions with large samples (13) in Figure 2b. It is not convincing to say that the deviation from the optimal model is not statistically significant, because this relates to variability in participants' data that their model should be explaining, not hiding behind. As shown in Figure S3b, the Difference heuristic seems to capture these data better than the optimal model across the range of sample sizes (less overestimation of slope for the smallest samples, less underestimation of slope for the largest samples). At the very least the authors should acknowledge this explicitly and give readers an intuition for why the optimal model nevertheless provides a better fit to the data in model comparisons. Otherwise it leaves the reader to puzzle this out for themselves. Relating to this, the authors choose an oddly indirect method to comparing model fits in the Supplementary information (p.4). Why fit a power law function to the modeled slopes, rather than comparing the model predictions directly to the data?

2) Some figure references in the manuscript haven't been updated to reflect changes made, e.g., references on line 413 and elsewhere to a Figure 7 that think I has been removed, and no

reference in the results text to the new Figure 8. The manuscript needs careful proofreading for this and other typos.

3) Line 428 states that “hierarchical integration offers a better explanation” but the comparison case isn’t clear. Better explanation than what alternative?

4) Line 510 states that “in these heuristics models evidence about the current context is averaged and not accumulated across trials”. This statement doesn’t seem to apply to the tally model. Please clarify, and explain why the predictions of the optimal and tally models are so different in the Fig. S10 plots. I was initially confused why the tally model should be insensitive to trial number. I think, but am not sure, this relates to the fact that the tally is used to estimate M rather than factor directly into the decision, in contrast to the tally model considered in Figure S7, so would like to see this point clarified.

REVIEWERS' COMMENTS:

Reviewer #2 (Remarks to the Author):

I have now read the paper and, more importantly, the rebuttal. My impression is not positive.

All three reviewers found the paper's key interesting point to be the claim that in an ecologically intuitive context, people can do probabilistic, sample-size sensitive judgements. All three reviewers also independently and unanimously ask for a control experiment in which the same task with abstract instructions is presented to participants and previous insensitivity to sample size is replicated.

The authors' response is to wriggle out of this straightforward request. They argue that their point is that probabilistic inference is "possible" and at least some subjects under some conditions can do it. They also acknowledge that others (eg De Martino et al 2015) have indeed demonstrated sample size sensitivity and that others (eg Purcell & Kiani) have demonstrated hierarchical probabilistic reasoning in humans. All of this puts the reader in a difficult position: the paper is now reduced to a very restricted claim that has been previously demonstrated several times. It also retracts the very claim that all three reviewers found interesting about the paper (ie the role of intuitive context).

To me, the paper is now much less interesting. I do not find the claims bold or controversial anymore and pretty much agree with reviewer 3 that, anyone who runs a probabilistic learning task with clear enough instructions and adequate practice will demonstrate the same result WITHOUT adding anything new to our understanding of the cognitive basis of probabilistic inference in human mind and brain.

We have tried our best to address the reviewer's comment, although we largely disagree. As required by the editor, we now make a clear statement about the limitation of the study: *"One clear limitation of our study is that it shows that humans can use reliability-based hierarchical integration of evidence, but it does not speak to the circumstances when this occurs."* (Discussion, lines 597-599). We also clearly state that some elements of the results had been shown in isolation in previous studies: *"In summary, while previous studies had already shown in isolation sample size sensitivity²⁹ and some form of hierarchical probabilistic reasoning^{1,7,8,15}, here the conjunction of both phenomena and the very detailed correspondence between human and optimal behaviors builds strong evidence for ubiquitous reliability-based integration of hierarchical information, even without extensive prior training on the task."* (lines 580-4).

Regarding the other criticisms of the study, our impression is that they are already discussed at length in the Discussion. We copy below the related excerpts.

- Regarding the fact that others had already reported hierarchical probabilistic reasoning in humans:
"Previous work has studied perception and decision making in similar hierarchical schemes like ours^{1,7,8,15}, but it has been difficult to independently modulate the reliability at both higher and lower hierarchical levels. For example, when using stimulus duration and stimulus strength as an indirect proxy to control reliability¹⁵, the

way these manipulations affect reliability depends on the specifics of the sensory system and sensory noise. In our task, uncertainty emerged not from sensory noise but from a hidden cause for stochastically generated stimuli, and the reliability of both levels could be controlled directly and independently through sample size, thus providing an objective measure of trial-to-trial reliability independent of the sensory system. Our task revealed that humans modulate their confidence not only based on the reliability of the currently observed sample, but also on the inferred reliability of the context which is itself a function of previous samples.”

- Regarding the fact the “anyone who runs a probabilistic learning task with clear enough instructions and adequate practice will demonstrate the same result”, we must state that sample size insensitivity reported in many previous studies is a clear sign that participants can repeatedly perform a probabilistic learning task and still not do it probabilistically. We also point to the point that practice most probably did not play any role in our findings as there was no informative feedback about the correct confidence report.

“In our task, for instance, learning calibrated confidence reports would require repeated exposure to the same sample together with supervising feedback about the actual latent variable (airplane majority). Even for very simple problems, the scarcity of such data makes this frequentist approach to uncertainty estimation practically difficult and thus un-ecological. As we did not provide supervising feedback, our participants presumably held accurate internal trial-by-trial representations of uncertainty^{38,39}.”

Reviewer #3 (Remarks to the Author):

Note: The editor asked me to evaluate the manuscript also in relation to Reviewer 1’s comments.

The authors have provided a very responsive revision of the manuscript, including several new analyses and plots that expand and reassure on important points. The clarification about the nature of feedback (or lack of it) provided to participants addresses most of my comments and those of Reviewer 1, and the new model fit in Fig. 8 is convincing. Altogether the manuscript makes a convincing argument about an important and timely issue. It would nevertheless benefit from attention to a few remaining minor issues:

1) In my first review I highlighted the deviation from model predictions with large samples (13) in Figure 2b. It is not convincing to say that the deviation from the optimal model is not statistically significant, because this relates to variability in participants’ data that their model should be explaining, not hiding behind. As shown in Figure S3b, the Difference heuristic seems to capture these data better than the optimal model across the range of sample sizes (less overestimation of slope for the smallest samples, less underestimation of slope for the largest samples). At the very least the authors should acknowledge this explicitly and give readers an intuition for why the optimal model nevertheless provides a better fit to the data in model comparisons. Otherwise it leaves the reader to puzzle this out for themselves. Relating to this, the authors choose an oddly indirect method to comparing model fits in the

Supplementary information (p.4). Why fit a power law function to the modeled slopes, rather than comparing the model predictions directly to the data?

We agree with the reviewer that the averaged data do not tell everything about how well a model accounts for the data as it hinders participant variability. As a minor point, we do not think that the Difference heuristic performs better than the optimal observer on this plot: there is indeed less underestimation for low sample size in the Difference heuristic, but more departure from sample size 9-11 (outside the confidence interval for the Difference heuristic). This is now acknowledged in the description of the figure (Supp Fig 3 Legend):

“Slopes in the difference model qualitatively matched that of participants. The goodness of fit of the difference model is roughly similar to that of the optimal model (figure 2b): underestimation of slope is larger in the optimal model for small sample sizes, but discrepancy with participant data is larger for sample size 9-11 in the difference model. Remember that this comparison is made on data averaged over participants. This motivated us to look for a single metric that can be estimated on each participant and allow more fine-grained comparison between the models.”

More importantly, we looked for a more principled way of comparing the two models. One is Bayesian model selection technique. The other was capturing the overall curve for each participant by a simple metric (allowing robust estimation for each participant): the exponent in the power law function. The reason we used this formula is because our analytical derivations (in Supplementary Methods) showed that the different models (optimal, ratio, difference) simply differed in that exponent (respectively 1, 0 and 0.5). This was confirmed by simulations (Supp Fig 3c), although the values do not exactly match the theoretical ones due to the distortion of responses. The model selection approach and power law approach both identified the optimal model as the model that better accounts for participant behavior. We agree that this was not very clearly motivated in the text, we have now added in Supplementary Methods: *“In summary, the models predicted a different exponent in the power law for the relation between sample size and slope: 0 for the ratio model (no dependence on sample size); 1 for the difference model (linear dependence); and 0.5 for the optimal model (sublinear dependence). Extracting the power law for each participant thus provides a single metric for sensitivity to sample size that can characterize whether each participant behaved more like the ratio, difference or optimal model.”*

2) Some figure references in the manuscript haven't been updated to reflect changes made, e.g., references on line 413 and elsewhere to a Figure 7 that think I has been removed, and no reference in the results text to the new Figure 8. The manuscript needs careful proofreading for this and other typos.

We inadvertently put Figure 6 instead of Figure 7 when we updated the figures (so Figure 6 was repeated), which obviously ruined all reference to the figure. We are very sorry for this and thank the reviewer for noticing the incongruity. This has now been corrected.

There was only one reference to the new Figure 8 in Discussion (line 623 of current version). The reference was indeed absent from the corresponding Result section; this has now been corrected (L478 of new manuscript).

Finally, we have made additional proofreading and hope there is no typo left.

3) Line 428 states that “hierarchical integration offers a better explanation” but the comparison case isn't clear. Better explanation than what alternative?

We agree that the sentence was not clear at all. We have removed reference to any comparison, as the sentence now reads (line 444): *“Overall, hierarchical integration offers a parsimonious explanation for context integration which does not require explicit memorization of previous samples after they are integrated into the context-level variable.”*

4) Line 510 states that “in these heuristics models evidence about the current context is averaged and not accumulated across trials”. This statement doesn’t seem to apply to the tally model. Please clarify, and explain why the predictions of the optimal and tally models are so different in the Fig. S10 plots. I was initially confused why the tally model should be insensitive to trial number. I think, but am not sure, this relates to the fact that the tally is used to estimate M rather than factor directly into the decision, in contrast to the tally model considered in Figure S7, so would like to see this point clarified.

We thank the reviewer for her/his interesting comment. In the tally model, the message is computed by pooling all the blue dots and red dots from previous trials in the block and computing the average proportion of blue dots over these tallies (see equation 11). This is exactly the ‘aligned cumulative proportion’ used as X axis in Supplementary Figure 10, so it is no wonder that the model shows no further modulation by trial position in this figure. To make it more explicit, the model calculates the same message $M=0.8$ if there was just one trial before with 4 blue dots and 1 red dot, or three trials in a row with 4 blue dots and 1 red dot each. This is why in the tally model, evidence is averaged and not accumulated. To make this more explicit to the reader, we have added reference in the main text to the equations for model evidence according to the different heuristic models.